# Knowledge-Guided Wasserstein Distributionally Robust Optimization

**Zitao Wang** [1]  **Ziyuan Wang** [2]  **Molei Liu** [* 3]  **Nian Si** [* 4]

## Abstract

Wasserstein Distributionally Robust Optimization (WDRO) is a principled framework for robust estimation under distributional uncertainty. However, its standard formulation can be overly conservative, particularly in small-sample regimes. We propose a novel knowledge-guided WDRO (KG-WDRO) framework for transfer learning, which adaptively incorporates multiple sources of external knowledge to improve generalization accuracy. Our method constructs smaller Wasserstein ambiguity sets by controlling the transportation along directions informed by the source knowledge. This strategy can alleviate perturbations on the predictive projection of the covariates and protect against information loss. Theoretically, we establish the equivalence between our WDRO formulation and the knowledge-guided shrinkage estimation based on collinear similarity, ensuring tractability and geometrizing the feasible set. This also reveals a novel and general interpretation for recent shrinkage-based transfer learning approaches from the perspective of distributional robustness. In addition, our framework can adjust for scaling differences in the regression models between the source and target and accommodates general types of regularization such as lasso and ridge. Extensive simulations demonstrate the superior performance and adaptivity of KG-WDRO in enhancing small-sample transfer learning.

*Equal contribution [1]Department of Statistics, Columbia University, New York, USA. [2]Department of Industrial Engineering and Management Sciences, Northwestern University, Evanston, USA. [3]Department of Biostatistics, Peking University Health Science Center; Beijing International Center for Mathematical Research, Peking University, Beijing, China. [4]Department of Industrial Engineering and Decision Analytics, Hong Kong University of Science and Technology, Hong Kong, China.. Correspondence to: Nian Si <niansi@ust.hk>, Molei Liu <moleiliu@bjmu.edu.cn>.

*Proceedings of the $42^{nd}$ International Conference on Machine Learning*, Vancouver, Canada. PMLR 267, 2025. Copyright 2025 by the author(s).

## 1. Introduction

Traditional machine learning methods or empirical risk minimization often suffer from overfitting and a lack of generalization power, particularly in high-dimensional and small-sample-size settings. In recent years, distributionally robust optimization (DRO) has emerged as a powerful framework for mitigating the effects of model misspecification and enhancing robustness in machine learning generalizations. Among various DRO formulations, Wasserstein-DRO (WDRO) gained more attention due to its tractability and generalizability. Specifically, in WDRO, one optimizes over worst-case distributions within an ambiguity set defined by a Wasserstein ball centered at an empirical measure.

However, one persistent challenge with WDRO is its tendency to be overly conservative, which can lead to suboptimal performance in practice as found in (Liu et al., 2024). In many real-world scenarios, prior knowledge can be leveraged to improve model performance and robustness. This transfer learning approach falls under the category of *Domain Adaptation*, which adapts models trained on a source domain to perform well on a related target domain with limited labeled data.

A key application is in clinical trials, where the binary outcome $Y \in \{0, 1\}$ indicates treatment success or failure, and the high-dimensional covariate $X$ encodes a patient's physical and health conditions along with treatment details. Data scarcity is common—especially for underrepresented populations. To address this, we leverage a classifier trained on a majority group (parameterized by $\theta$) as a reference to estimate a classifier for the minority group (parameterized by $\beta$). This knowledge-guided transfer learning reduces uncertainty by anchoring the search for $\beta$ in the direction of $\theta$. In such a context, transfer learning has proven to be a versatile approach for improving performance on a target task. *Despite its successes, the integration of prior knowledge into WDRO frameworks has remained an open question.*

In this work, we introduce Knowledge-Guided Wasserstein Distributionally Robust Optimization (KG-WDRO), a novel framework that adapts the Wasserstein ambiguity set using external knowledge (parameters). We assume access to prior predictors of pre-trained models, which can guide the predictive model in the target dataset. By constraining the transport cost along directions informed by prior

knowledge, our approach addresses the conservativeness of vanilla WDRO while preserving robustness. Intuitively, this strategy allows the model to focus its uncertainty on regions where prior knowledge is less reliable, effectively robustify knowledge-guided generalization.

## 1.1. Related Works

### 1.1.1. WASSERSTEIN DRO

Wasserstein DRO has recently garnered significant attention due to its tractability (Mohajerin Esfahani & Kuhn, 2018; Blanchet & Murthy, 2019; Gao & Kleywegt, 2023) and generalizability (Blanchet et al., 2019a; Gao et al., 2022). Notably, Blanchet et al. (2019a) and Gao et al. (2022) demonstrate that Wasserstein DRO with mean square loss is equivalent to the square root lasso (Belloni et al., 2011). Similarly, Shafieezadeh-Abadeh et al. (2015; 2019); Blanchet et al. (2019a); Gao et al. (2022) establish that Wasserstein DRO with logistic loss and hinge loss corresponds to their regularized counterparts. Moreover, the statistical properties of the WDRO estimator have also been investigated in (Blanchet et al., 2021; 2022; Gao, 2023). However, leveraging external knowledge in Wasserstein DRO has been an open problem.

### 1.1.2. TRANSFER LEARNING

Improving prediction accuracy for target populations by integrating diverse source datasets has driven methodological advances in transfer learning. Contemporary approaches aim to address challenges including distributional heterogeneity and limited labeled target data. A common assumption is that the target outcome model aligns partially with source models, enabling knowledge transfer. For example, recent frameworks employ selective parameter reduction to identify transferable sources and sparse or ridge shrinkage to leverage their knowledge (Bastani, 2020; Li et al., 2021; Tian & Feng, 2023). Subsequent works tackle covariate distribution mismatches and semi-supervised scenarios, enhancing robustness when labeled target data is scarce (Cai et al., 2024; He et al., 2024; Zhou et al., 2024). Further innovations include geometric or profile-based adaptations, where the target model is represented as a weighted combination of source coefficients (Gu et al., 2024; Lin et al., 2024).

## 1.2. Our Contribution

Our contributions are fourfold. **Framework:** We introduce KG-WDRO, a principled and flexible framework that integrates prior knowledge into WDRO for linear regression and binary classification. This framework mitigates the conservativeness of standard WDRO, enables automated covariate scaling adjustments, and prevents negative transfer. **Theory:** We establish the equivalence between KG-

WDRO and shrinkage-based estimation methods, offering a novel perspective that unifies and interprets a broad range of knowledge transfer learning approaches through the lens of distributional robustness. Table 1 provides an overview of them, highlighting their key capabilities and advantages and comparing them with our framework. **Technicalities:** Leveraging Toland's Duality (Theorem F.1), we reformulate the innermost maximization in WDRO's strong duality (Proposition 2.1) into a univariate optimization problem (Toland's Duality). This reformulation enhances tractability while accommodating more general cost functions. **Empirical Validation:** Through extensive experiments, we demonstrate the effectiveness of KG-WDRO in improving small-sample transfer learning.

Below is an overview of our main results for the linear regression case.

*Example* 1. Suppose $\theta$ is an accessible prior predictor for a linear model parameterized with $\beta$. We show that the shrinkage-based transfer-learning regression problem, which estimates a target predictor $\beta$ by solving

$$\inf_{\beta, \kappa \in \mathbb{R}} \|\mathbf{y} - \mathbf{X}\beta\|_2 + \sqrt{\delta}\|\beta - \kappa\theta\|_p,$$

can be interpreted as a Wasserstein distributionally robust optimization (WDRO) problem of the form (WDRO), where the loss function is least squares, $\ell(X, Y; \beta) = (Y - \beta^\intercal X)^2$, and the ambiguity set $\mathcal{B}_\delta(\mathbb{P}_N; c_{2,\infty})$ is defined as a ball around the empirical measure. The cost function $c_{2,\infty}$ augments the standard transport cost by the constraint $x^\intercal \theta = u^\intercal \theta$ so that

$$
\begin{aligned}
&c_{2,\infty}\big((x,y),(u,v)\big) \\
=&\|x - u\|_q^2 + \infty \cdot |y - v| + \infty \cdot |(x - u)^\intercal \theta|.
\end{aligned}
$$

This establishes a distributionally robust optimization (DRO) perspective on a broad class of transfer-learning methods as will be discussed in Section 3.

## 1.3. Notations & Organizations

We summarize the mathematical notations used in this work. The positive integers $N$, $M$, and $d$ denote, respectively, the target sample size, the number of sources, and the dimension of the support of the covariate $X$. The integers $p$ and $q \in [1, \infty]$ are reserved for pairs of Hölder conjugates, satisfying $p^{-1} + q^{-1} = 1$ for $p, q \in (1, \infty)$, as well as the pair 1 and $\infty$. For a distribution $\mathbb{P}$ supported on the Euclidean space $\mathbb{R}^d$, we use $\mathbb{P}_N$ to denote the empirical measure of $\mathbb{P}$ with sample size $N$. In modeling the target-covariate relationship, the distribution is often factorized as $\mathbb{P} = \mathbb{P}^{Y|X} \times \mathbb{P}^X$. For a vector $v \in \mathbb{R}^d$, $\|v\|_p$ denotes the $p$-norm, where $p \in [1, \infty]$, and $v^\intercal$ denote the transpose of $v$. For any two vectors $u, v \in \mathbb{R}^d$, the notation $\cos(u, v)$ denote the cosine of the angle between $u$ and $v$, calculated by $\cos(u, v)\|u\|_2\|v\|_2 =$

*Table 1.* Overview of recent transfer learning techniques. Each column represents a key capability: **Ridge-type / Lasso-type** – Regularization type used; **Scale Adjustment** – Robustness against feature-wise scaling; **Continuous outcome / Binary outcome** – Supports regression or classification; **Partial Transfer** – Selections of prior knowledge; **Multi-Source ensemble** – Profiles on multiple prior knowledges.

| Methods | Ridge -Type | Lasso -Type | Scale Adjustment | Continuous Outcome | Binary Outcome | Partial Transfer | Multi-Source Ensemble |
|---|:---:|:---:|:---:|:---:|:---:|:---:|:---:|
| KG-WDRO | ✓ | ✓ | ✓ | ✓ | ✓ | ✓ | ✓ |
| Bastani (2020) | ✓ | | | ✓ | | | |
| Li et al. (2021) | | ✓ | | ✓ | | | |
| Tian & Feng (2023) | | ✓ | | ✓ | ✓ | | |
| Gu et al. (2024) | ✓ | | ✓ | ✓ | | | ✓ |
| Lin et al. (2024) | | ✓ | ✓ | ✓ | | | ✓ |

$u^\top v$. All vectors are assumed to be column vectors. Other specialized notations are defined in context as needed.

The remainder of the paper is organized as follows. Section 2 provides a review of the WDRO framework, including the strong duality result. In Section 3, we introduce our KG-WDRO framework and demonstrate its equivalence to shrinkage-based estimations in both linear regression and binary classification. Section 4 presents comprehensive results from our numerical simulations. All proofs and detailed descriptions of the numerical simulation setups are provided in the appendix.

## 2. Preliminaries

We first begin with a short overview of the distributionally robust framework on statistical learning.

### 2.1. Optimal Transport Cost

Let $\mathbb{P}$ and $\mathbb{Q}$ denote two probability distributions supported on $\mathbb{R}^d$, and we use $\mathcal{P}(\mathbb{R}^d \times \mathbb{R}^d)$ to label the set of all probability measures on the product space $\mathbb{R}^d \times \mathbb{R}^d$. We say that an element $\pi \in \mathcal{P}(\mathbb{R}^d \times \mathbb{R}^d)$ has first marginal $\mathbb{P}$ and second marginal $\mathbb{Q}$ if

$$\pi(A \times \mathbb{R}^d) = \mathbb{P}(A), \;\; \pi(\mathbb{R}^d \times B) = \mathbb{Q}(B),$$

for all Borel measurable sets $A, B \in \mathbb{R}^d$. The class of all such measures $\pi$ is collected as $\Pi(\mathbb{P}, \mathbb{Q})$, and is called the set of *transport plans*, which is always non-empty. Choose a non-negative, lower semi-continuous function $c : \mathbb{R}^d \times \mathbb{R}^d \to [0, \infty]$ such that $c(u, v) = 0$ whenever $u = v$, then the *Kantorovich's formulation* of optimal transport is defined as

$$\mathcal{D}_c(\mathbb{P}, \mathbb{Q}) := \inf_{\pi \in \Pi(\mathbb{P}, \mathbb{Q})} \mathbb{E}_\pi [c(U, V)].$$

It is well-known that (Villani, 2009, Theorem 4.1) there exists an optimal coupling $\pi^\dagger$ that solves the Kantorovich's problem $\inf_{\pi \in \Pi(\mathbb{P}, \mathbb{Q})} \mathbb{E}_\pi [c(U, V)]$. Intuitively, we may

think of the value $c(u, v)$ as the cost of transferring one unit of mass from $u \in \mathbb{R}^d$ to $v \in \mathbb{R}^d$, then $\mathbb{E}_\pi[c(U, V)]$ gives the average cost of transferring under the plan $\pi$. The *optimal transport cost* $\mathcal{D}_c(\mathbb{P}, \mathbb{Q})$ gives a measure of discrepancy between probability distributions on $\mathbb{R}^d$.

If $c(u, v)$ defines a metric on $\mathbb{R}^d$, then for any $p \in [1, \infty)$ the optimal transport cost,

$$\mathcal{D}_c^{1/p}(\mathbb{P}, \mathbb{Q}) := \left( \inf_{\pi \in \Pi(\mathbb{P}, \mathbb{Q})} \mathbb{E}_\pi \left[ c(U, V)^p \right] \right)^{1/p},$$

defines a metric between probability distributions and metrizes weak convergence under moment assumptions. It is called the $p$-Wasserstein distance. We direct the interested readers to (Villani, 2009, Chapter 6) for more details. It is worth mentioning that none of our judiciously chosen cost functions qualify as metrics on the support of the data.

### 2.2. Distributionally Robust Optimization

In standard statistical learning framework, one generally assumes that the target-covariate pair $(X, Y) \in \mathbb{R}^d \times \mathbb{R} \cong \mathbb{R}^{d+1}$ follows a data-generating distribution $\mathbb{P} := \mathbb{P}_{X,Y}$ on the support $\mathbb{R}^{d+1}$. One then seeks to find a 'best' parameter $\beta$ that relates $Y$ to $X$ through a parameterized model by solving the stochastic optimization,

$$\inf_\beta \mathbb{E}_\mathbb{P} \left[ \ell(X, Y; \beta) \right]. \tag{SO}$$

The *loss function* $\ell(x, y; \beta)$ provides a quantification of the goodness-of-fit in the parameter $\beta$ given the realized observation $(x, y)$. Since only samples $\{(x_i, y_i)\}_{i=1,\dots,N}$ are observed, we can typically only solve the empirical objective,

$$\inf_\beta \mathbb{E}_{\mathbb{P}_N} [\ell(X, Y; \beta)] = \inf_\beta \frac{1}{N} \sum_{i=1}^N \ell(x_i, y_i; \beta). \tag{ERM}$$

Therefore the distribution $\mathbb{P}$ that underlies the data-generating mechanism is uncertain to the decision-maker.

This motivated the *distributionally robust optimization* (DRO) framework, which entails solving the following min-imax stochastic program:

$$\inf_{\beta} \sup_{\mathbb{P} \in \mathcal{P}_{\text{amb}}} \mathbb{E}_{\mathbb{P}}[\ell(X, Y; \beta)], \qquad \text{(DRO)}$$

where the *ambiguity set* $\mathcal{P}_{\text{amb}}$ represents a class of probability measures supported on $\mathbb{R}^{d+1}$ that are candidates to the true data-generating distributions. In *Wasserstein*-DRO, the ambiguity set is constructed by forming a '$\delta$-ball' around the canonical empirical measure $\mathbb{P}_N$ associated to the decision-maker-defined transport cost $c$, i.e. we let the ambiguity set $\mathcal{P}_{\text{amb}}$ be chosen as:

$$\mathcal{B}_{\delta}(\mathbb{P}_N; c)$$
$$:= \{\mathbb{P} \in \mathcal{P}(\mathbb{R}^{d+1}) | \mathcal{D}_c(\mathbb{P}, \mathbb{P}_N) \leq \delta\}. \qquad \text{(WDRO)}$$

This ambiguity set captures probability measures that are close to the observed empirical measure in the transport cost $\mathcal{D}_c$, which may be taken as a class of candidates of measures perturbed from $\mathbb{P}_N$. The solution $\beta_{\text{DRO}}$ to (DRO) that solves the worst case expected loss should perform well over the entire set of perturbations in the ambiguity set. This is in contrast to $\beta_{\text{ERM}}$ that solves (ERM) only performs well on the training samples. This adds a robustness layer to the WDRO problem (WDRO). For a comprehensive overview of different constructions of ambiguity sets, we direct the interested reader to (Kuhn et al., 2024, Section 2).

## 2.3. Strong Duality of Wasserstein DRO

The Wasserstein DRO problem involves an inner maximization over an infinite-dimensional set, which appears computationally intractable. However, the distribution $\mathbb{P}_n$ is discrete, strong duality of the Wasserstein DRO reformulates it as a simple univariate optimization.

**Proposition 2.1** (Strong Duality, (Blanchet et al., 2019a, Proposition 1)). *Let $c : \mathbb{R}^{d+1} \times \mathbb{R}^{d+1} \to [0, \infty]$ be a lower semi-continuous cost function satisfying $c\big((x, y), (u, v)\big) = 0$ whenever $(x, y) = (u, v)$. Then the distributionally robust regression problem*

$$\inf_{\beta \in \mathbb{R}^d} \sup_{\mathbb{P}: \mathcal{B}_{\delta}(\mathbb{P}_N)} \mathbb{E}_{\mathbb{P}} \left[ \ell(X, Y; \beta) \right],$$

*is equivalent to,*

$$\inf_{\beta \in \mathbb{R}^d} \inf_{\gamma \geq 0} \left\{ \gamma \delta + \frac{1}{n} \sum_{i=1}^{N} \phi_{\gamma}(x_i, y_i; \beta) \right\},$$

*where $\phi_{\gamma}(x_i, y_i; \beta)$ is given by,*

$$\sup_{(u,v) \in \mathbb{R}^{d+1}} \left\{ \ell(u, v; \beta) - \gamma c\big((u, v), (x_i, y_i)\big) \right\}.$$

For more general results, see (Blanchet & Murthy, 2019, Theorem 1) and (Gao et al., 2022, Section 2). The exchangeability of sup and inf in Wasserstein-DRO is also established by (Blanchet et al., 2019a, Lemma 1).

## 3. Knowledge-Guided Wasserstein DRO

In this section, we propose new cost functions for the Wasserstein DRO framework that leverage prior knowledge for transfer learning. For linear regression and binary classification, these cost functions act as regularizers, encouraging collinearity with prior knowledge.

### 3.1. Knowledge-Guided Transport Cost

It is shown in (Blanchet et al., 2019a, Theorem 1) that using the squared $q$-norm on the covariates as the cost function

$$c_2\big((x, y), (u, v)\big) = \|x - u\|_q^2 + \infty \cdot |y - v|, \qquad (1)$$

equates Wasserstein distributionally robust linear regression with $p$-norm regularization on the root mean squared error (RMSE). The cost function $c_2$ perturbs only the observed covariates $\{x_i\}_{i=1}^N$, while keeping the observed targets $\{y_i\}_{i=1}^N$ fixed. Keeping the observed target $Y$ as fixed often leads to more mathematically tractable reformulation, another intuition is that we trust the mechanism by which the target $Y$ is generated once $X$ is known.

In the presence of prior knowledge $\theta$ that may aid in inferring $\beta$, we aim to control the extent of perturbation along the direction of $\theta$.

Specifically, we constrain the size of the prediction discrepancy $\theta^{\mathsf{T}} x - \theta^{\mathsf{T}} u = \theta^{\mathsf{T}} \Delta$, where $\Delta := x - u$. To achieve this goal, we henceforth augment the cost function $c_2$ with an additional penalty term that accounts for the size of the perturbation in the direction of $\theta$:

$$c_{2,\lambda}\big((x, y), (u, v)\big)$$
$$= \|\Delta\|_q^2 + \infty \cdot |y - v| + \lambda h(|\theta^{\mathsf{T}} \Delta|), \qquad (2)$$

where $\lambda > 0$ and $h(x) : \mathbb{R} \to \mathbb{R}^+ \cup \{0\}$ is a non-negative, monotone increasing function of $|x|$ such that $h(0) = 0$. Recall that in the cost function $c_2(\cdot)$, the targets $y$ remain fixed. Intuitively, the new cost function (2) encourages the Wasserstein ambiguity set to include distributions whose marginals in $X$ generate predictions that align with the data based on the prior predictor $\theta$. The parameter $\lambda$ controls the level of confidence in the prior knowledge. We call this kind of cost functions *knowledge-guided*. Since $c_{2,\lambda}$ upper bounds the cost function $c_2$, we have $\mathcal{B}_{\delta}(\mathbb{P}_N^X; c_{2,\lambda_2}) \subseteq \mathcal{B}_{\delta}(\mathbb{P}_N^X; c_{2,\lambda_1}) \subseteq \mathcal{B}_{\delta}(\mathbb{P}_N^X; c_2)$ whenever $\lambda_2 > \lambda_1$.

The corresponding optimal transport problem given by:

$$\inf_{\pi \in \Pi(\mathbb{Q}^X, \mathbb{P}_N^X)} \mathbb{E}_{\pi}[c_{2,\lambda}(X, U)],$$

can also be expressed as:

$$\inf_{\pi \in \Pi(\mathbb{Q}^X, \mathbb{P}_N^X)} \mathbb{E}_{\pi}[c_2(X, U)] + \lambda \mathbb{E}_{\pi}[h(|\theta^{\mathsf{T}} \Delta|)].$$

This formulation regularizes the original optimal transport problem by penalizing large values of the expectation $\mathbb{E}_\pi[h(|\theta^\mathsf{T}\Delta|)]$.

For any user-defined function $h$ that measures the discrepancy in generalization with respect to the prior knowledge $\theta$, we refer to it as *weak-transferring* of knowledge if $\lambda < +\infty$, and *strong-transferring* of knowledge if $\lambda = +\infty$. In the case of strong-transferring, to ensure the finiteness of the optimal transport problem, the minimizing transport plan $\pi^\dagger$ must satisfy the orthogonality condition $\theta^\mathsf{T}\Delta = 0$, $\pi^\dagger$-almost surely. Consequently, the value of $\theta^\mathsf{T}X$ remains unchanged after perturbing $\mathbb{P}_N^X$ within $\mathcal{B}_\delta(\mathbb{P}_N^X; c_{2,\infty})$. As a result, this should promote $\beta_{\mathrm{DRO}} \to \theta$ as $\delta \to \infty$. Indeed, we have the following proposition on the bound of the minimax objective with strong transferring.

**Proposition 3.1.** *Let $\ell(X, Y; \beta) = (Y - \beta^\mathsf{T}X)^2$ denote the least square loss, then*

$$\inf_{\beta\in\mathbb{R}^d} \sup_{\mathbb{P}:\mathcal{B}_\delta(\mathbb{P}_N;c_{2,\infty})} \mathbb{E}_\mathbb{P}\big[(Y - \beta^\mathsf{T}X)^2\big]$$
$$\leq \inf_{\alpha\in\mathbb{R}} \mathbb{E}_{\mathbb{P}_N^*}\big[(Y - (\alpha\theta)^\mathsf{T}X)^2\big].$$

Thus, the minimax optimizer under knowledge guidance from $\theta$ achieves out-of-sample performance that is at least as good as the in-sample performance of the naive domain adapter $\widehat{\alpha}_N\theta$. A similar statement applies to the binary classification settings as discussed in Theorem 3.6.

*Remark* 3.2. The above framework extends to incorporate multi-sites prior knowledge, meaning that instead of a single prior knowledge coefficient $\theta_1$, we consider a set of coefficients $\{\theta_1, \theta_2, \ldots, \theta_M\}$. Let $\Theta := \mathrm{span}\{\theta_1, \theta_2, \ldots, \theta_M\}$ represent the linear span of these prior knowledge coefficients. In the case of strong-transferring, we must ensure that $\mathrm{rank}(\Theta) < d$; otherwise, the set of orthogonality conditions $\{\theta_m^\mathsf{T}\Delta = 0; m \in [M]\}$ would imply that the perturbation $\Delta$ is identically zero ($\Delta = \mathbf{0}$). This would render the ambiguity set redundant and reduce the WDRO problem (WDRO) to the ERM problem (ERM). This result is confirmed by the statements of Theorems 3.3 and 3.6.

### 3.2. Linear Regression

We begin by examining the WDRO problem (WDRO) for linear regression within the strong-transferring domain. Following this, we present a specific case within the weak-transferring domain. Let $\Theta := \mathrm{span}\{\theta_1, \ldots, \theta_M\}$ represent the linear span of the prior knowledge.

#### 3.2.1. STRONG-TRANSFERRING

Define the cost function $c_{2,\infty}\big((x,y),(u,v)\big) := \|x - u\|_q^2 + \infty \cdot |y - v| + \infty \cdot |\theta_1^\mathsf{T}x - \theta_1^\mathsf{T}u| + \ldots + \infty \cdot |\theta_M^\mathsf{T}x - \theta_M^\mathsf{T}u|$, and for a set of observed samples $\{(x_i, y_i)\}_{i\in[N]}$, we use $\mathrm{MSE}_N(\beta) := N^{-1}\sum_{i=1}^N (y_i - \beta^\mathsf{T}x_i)^2$. Without making

any additional distributional assumptions on $(X, Y)$, we obtain the following finite-dimensional representation.

**Theorem 3.3** (Linear Regression with Strong-Transferring). *Consider the least-squared loss $\ell(X, Y; \beta) = (Y - \beta^\mathsf{T}X)^2$, then for any $q \in [1, \infty]$ we have*

$$\inf_{\beta\in\mathbb{R}^d} \sup_{\mathbb{P}:\mathcal{B}_\delta(\mathbb{P}_N;c_{2,\infty})} \mathbb{E}_\mathbb{P}\big[(Y - \beta^\mathsf{T}X)^2\big]$$
$$= \inf_{\beta\in\mathbb{R}^d, \vartheta\in\Theta} \left(\sqrt{\mathrm{MSE}_N(\beta)} + \sqrt{\delta}\|\beta - \vartheta\|_p\right)^2,$$

*where $p$ is such that $p^{-1} + q^{-1} = 1$.*

From the above result, we observe that the knowledge-guided WDRO problem for linear regression is equivalent to regularizing the RMSE with a $p$-norm distance to the linear span $\Theta$. The regularization parameter is entirely determined by the size (or radius) of the Wasserstein ambiguity set. Importantly, the penalty term focuses on the collinearity with the prior knowledge rather than their algebraic difference or angular proximity.

Consider the case when there is only a single prior knowledge $\theta_1$, the penalty term does not constrain the solution $\beta_{\mathrm{DRO}}$ to be close to $\theta_1$, but rather to $\kappa \cdot \theta_1$ for some $\kappa \in \mathbb{R}$ to be optimized. Consequently, this knowledge transfer automatically robustify solution against scaling of covariates. Furthermore, it can prevent negative transfer by adapting its sign to be positively correlated with $\beta^*$, which is the solution to population objective (SO). When $\delta \to \infty$, the penalty term becomes dominant, forcing $\beta$ to lie in $\Theta$ for any $p \geq 1$. This reduces the WDRO problem to a simple constrained regression problem,

$$\inf_{\beta\in\Theta} \mathrm{MSE}_N(\beta),$$

reflecting the complete reliance on the prior knowledge and prevents excessive shrinkage towards the null estimator.

*Remark* 3.4. We now discuss two special cases of the penalty term, $p = 2$ (ridge-type regularization) and $p = 1$ (lasso-type regularization). For simplicity, we consider the case of a single prior knowledge vector $\theta$.

**Ridge-type.** The penalty term can be explicitly calculated as

$$\min_{\kappa\in\mathbb{R}} \|\beta - \kappa\theta\|_2 = \left\|\beta - \frac{\beta^\mathsf{T}\theta}{\|\theta\|_2^2}\theta\right\|_2 = \|\beta^{\perp\theta}\|_2,$$

where $\beta^{\perp\theta}$ is the component of $\beta$ orthogonal to $\theta$. This penalty term shrinks distance to the line in the direction of $\theta$. Furthermore, note that

$$\|\beta^{\perp\theta}\|_2 = \|\beta\|_2 \sin(\beta, \theta) = \|\beta\|_2\sqrt{1 - \cos^2(\beta, \theta)},$$

which represents a trade-off between the magnitude of $\beta$ and its angular proximity to the prior knowledge $\theta$. This

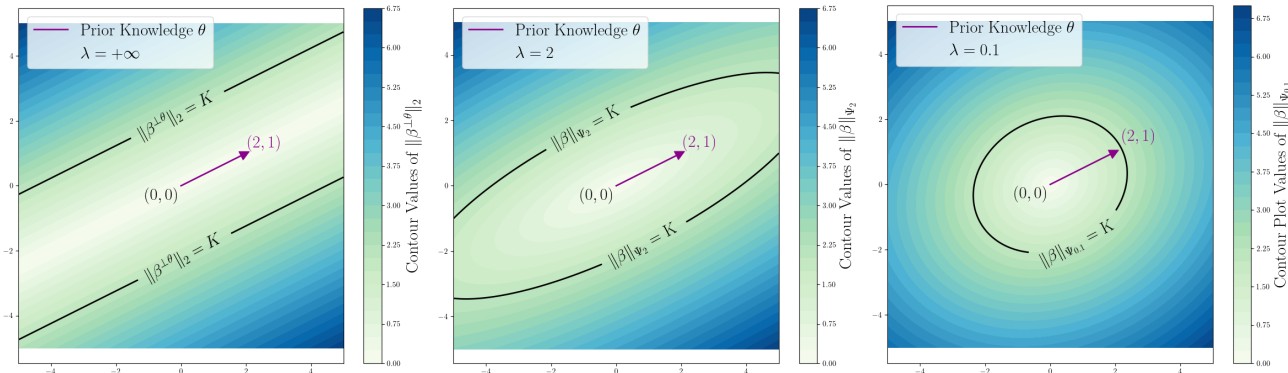

*Figure 1.* The two-dimensional contour plots of the regularization term in Theorem 3.3 and Theorem 3.5 with $\lambda$ ranging from $+\infty$ to 2 to 0.1. The prior knowledge parameter is taken as $\theta = (2,1)^\top$. The area between the black contours constitute a feasibility set of the regularization term when written in its equivalent constraint form. The feasibility set shrinks in the direction of $\theta$, to a circle of radius $K$ when $\lambda \to 0$ from above.

trade-off is illustrated in the leftmost figure of Fig.1, drawing the feasibility set of the regularization as a constraint. This regularization is closely related but different to the one proposed in (Gu et al., 2024), where they penalize large values of a computational relaxation of $\sin(\beta, \theta)$.

**Lasso-type.** When the prior knowledge $\theta$ is sparse, the penalty term $\min_\kappa \|\beta - \kappa\theta\|_1$ promotes sparse representation learning. Consider a simple example where the dimension is $d = 3$ and $\theta = (1,0,0)^\top$. In this case, we have:

$$\min_\kappa \|\beta - \kappa\theta\|_1 = \min_\kappa \left(|\beta_1 - \kappa| + |\beta_2| + |\beta_3|\right)$$
$$= |\beta_2| + |\beta_3| =: \|\beta_{-1}\|_1,$$

where $\beta_{-1} = (\beta_2, \beta_3)^\top$. This formulation enforces sparsity only on the last two components of $\beta$, reflecting the sparsity pattern of $\theta$.

### 3.2.2. WEAK TRANSFERRING

For the special case of $q = p = 2$, we define the weak-transferring cost function $c_{2,\lambda}\left((x,y),(u,v)\right) = \|x - u\|_2^2 + \lambda(\theta^\top x - \theta^\top u)^2 + \infty \cdot |y - v|$ with $0 < \lambda < +\infty$. Here, we select $h(x) = x^2$ as the user-defined function on controlling the size of perturbation in $\theta$. For simplicity, we consider a single prior knowledge vector $\theta$ in this setup. This result can be straightforwardly extended to a multi-source setup with different values of $\lambda$'s.

**Theorem 3.5** (Linear Regression with Weak Transferring). *Consider the least-squared loss $\ell(X,Y;\beta) = (Y - \beta^\top X)^2$, then for $p = q = 2$ we have*

$$\inf_{\beta \in \mathbb{R}^d} \sup_{\mathbb{P}:\mathcal{B}_\delta(\mathbb{P}_N;c_{2,\lambda})} \mathbb{E}_\mathbb{P}\left[(Y - \beta^\top X)^2\right]$$
$$= \inf_{\beta \in \mathbb{R}^d} \left(\sqrt{\mathrm{MSE}_N(\beta)} + \sqrt{\delta}\, \|\beta\|_{\Psi_{\lambda^{-1}}}\right)^2.$$

*With $\Psi_\lambda = I_d - \dfrac{1}{\|\theta\|_2^2 + \lambda}\theta\theta^\top$ and $\|\beta\|_{\Psi_\lambda}^2 = \beta^\top \Psi_\lambda \beta$.*

Write $P_\lambda = \theta\theta^\top/(\|\theta\|_2^2 + \lambda)$, we note that as $\lambda \to \infty$, we have $P_{\lambda^{-1}} \to P_0 = \theta\theta^\top/\|\theta\|_2^2$ recovering the projection matrix onto the prior knowledge $\theta$. Consequently, $\|\beta\|_{\Psi_{\lambda^{-1}}} \to \|\beta^{\perp\theta}\|_2$.

We observe that the action

$$P_{\lambda^{-1}}\beta = \frac{\beta^\top\theta}{\|\theta\|_2^2 + \lambda^{-1}}\theta$$

is exactly the ridge regression of $\beta$ onto $\theta$ with a regularization parameter $\lambda^{-1}$. Thus, the finiteness of $\lambda$, which can reflect a caution in the prior knowledge $\theta$, induces a shrinkage effect on the component of $\beta$ explainable by $\theta$ in the dot product geometry. Since $\Psi_\lambda \succ I_d - P$, we have $\|\beta\|_{\Psi_{\lambda^{-1}}} > \|\beta^{\perp\theta}\|_2$ for any finite $\lambda > 0$, this implies the inclusion of feasibility set

$$\{\beta : \|\beta\|_{\Psi_{\lambda^{-1}}} \leq K\} \subset \{\beta : \|\beta^{\perp\theta}\|_2 \leq K\},$$

as plotted in Fig.1 for an illustration on $\mathbb{R}^2$. The contour $\{\beta \in \mathbb{R}^2 : \|\beta\|_{\Psi_{\lambda^{-1}}} = K\}$ forms an ellipse centered around the origin $\mathbf{0}$. The ellipse has a major axis of half length $K\sqrt{\dfrac{\|\theta\|_2^2 + \lambda^{-1}}{\lambda^{-1}}}$ aligned with the direction of $\theta$, and a minor axis with half-length $K$ aligned with the direction of $\theta^\perp$. As $\lambda \to 0$, representing no-confidence in $\theta$, the half-length of the major axis converges to $K$, resulting in a perfect circle as in ridge regression.

The two-dimensional hyper-parameters $(\delta, \lambda^{-1})$ enable the use of data-driven methods, such as grid-search cross-validation, for hyper-parameter tuning. Unlike the strong-transferring domain, the inclusion of $\lambda^{-1}$ allows the data to self-determine the informativeness of the source samples.

## 3.3. Binary Classifications

In this section, we focus on the context of binary classification, where the goal is to predict the discrete label $Y \in \{-1, 1\}$ based on the covariates $X \in \mathbb{R}^d$. Unlike the previous section, we use the $q$-norm, rather than its square, to account for distributional ambiguity in the covariate distribution. Define the strong-transferring cost function $c_{1,\infty}\big((x,y),(u,v)\big) := \|x - u\|_q + \infty \cdot |y - v| + \infty \cdot |\theta_1^\mathsf{T} x - \theta_1^\mathsf{T} u| + \ldots + \infty \cdot |\theta_M^\mathsf{T} x - \theta_M^\mathsf{T} u|$. We consider two loss functions here. The **logistic loss** function is given by

$$\ell(X, Y; \beta) = \log\left(1 + e^{-Y\beta^\mathsf{T} X}\right),$$

which is the negative log-likelihood of the model that postulates

$$\log \frac{\mathbb{P}(Y = 1 | X = x)}{\mathbb{P}(Y = -1 | X = x)} = \beta^{*\mathsf{T}} x.$$

The **hinge loss** is given by

$$\ell(X, Y; \beta) = (1 - Y\beta^\mathsf{T} X)^+,$$

which is typically used for training classifiers that look for 'maximum-margins' in class boundaries, most notably *support vector machines*.

Suppose $Y \in \{-1, 1\}$ is binary and without any distributional assumptions on $X$, we have the following result which recovers regularized logistic regressions and support vector machines.

**Theorem 3.6** (Binary Classification with Strong Transferring)**.** *Suppose the loss function $\ell(X, Y; \beta)$ is either the logistic loss $\log\left(1 + e^{-Y\beta^\mathsf{T} X}\right)$ or the hinge loss $(1 - Y\beta^\mathsf{T} X)^+$, then for any $q \in [1, \infty]$ we have*

$$\inf_{\beta \in \mathbb{R}^d} \sup_{\mathbb{P}: \mathcal{B}_\delta(\mathbb{P}_N; c_{1,\infty})} \mathbb{E}_\mathbb{P}\left[\ell(X, Y; \beta)\right]$$

$$= \inf_{\beta \in \mathbb{R}^d, \vartheta \in \Theta} \frac{1}{N} \sum_{i=1}^N \ell(x_i, y_i; \beta) + \delta\|\beta - \vartheta\|_p,$$

*where $p$ is such that $p^{-1} + q^{-1} = 1$.*

## 3.4. Sub-Coefficient-Vector Transferring

In this subsection, we generalize the statements of Theorems 3.3 and 3.6 for $p = 2$ to arbitrary norms induced by positive-definite quadratic forms. Let $\Lambda \in \mathbb{R}^{d \times d}$ be a positive-definite symmetric matrix. The norm $\|x\|_\Lambda = \sqrt{x^\mathsf{T} \Lambda x}$ induces a metric on $\mathbb{R}^d$, defined as $d_\Lambda(x, u) = \|x - u\|_\Lambda$, known as the *Mahalanobis distance*. Since $\Lambda$ is positive definite, it admits a decomposition $\Lambda = \Gamma^\mathsf{T} \Gamma$ with $\Gamma$ invertible, and the norm $\|x\|_\Lambda = \|\Gamma x\|_2$ measures length in the geometry distorted by $\Gamma$. By (Blanchet et al., 2019b, Lemma 1), the dual norm of $\| \cdot \|_\Lambda$ is $\| \cdot \|_{\Lambda^{-1}}$. Using Proposition E.6,

the statements of Theorems 3.3 and 3.6 can be easily generalized. Define the space of positive-definite symmetric matrices as $\mathbb{S}_+^{d \times d}$ and the cost function: $c_{2,\infty}^\Lambda\big((x,y),(u,v)\big) := \|x - u\|_\Lambda^2 + \infty \cdot |y - v| + \infty \cdot \sum_{m=1}^M |\theta_m^\mathsf{T} x - \theta_m^\mathsf{T} u|$.

**Corollary 3.7** (Theorem 3.3)**.** *For the least-squares loss $\ell(X, Y; \beta) = (Y - \beta^\mathsf{T} X)^2$ and any $\Lambda \in \mathbb{S}_+^{d \times d}$:*

$$\inf_{\beta \in \mathbb{R}^d} \sup_{\mathbb{P}: \mathcal{B}_\delta(\mathbb{P}_N; c_{2,\infty}^\Lambda)} \mathbb{E}_\mathbb{P}\left[(Y - \beta^\mathsf{T} X)^2\right]$$

$$= \inf_{\beta \in \mathbb{R}^d, \vartheta \in \Theta} \left(\sqrt{\mathrm{MSE}_N(\beta)} + \sqrt{\delta}\|\beta - \vartheta\|_{\Lambda^{-1}}\right)^2.$$

This formulation enables the use of metric learning methods to determine $\Lambda$ directly from the data, as detailed in (Blanchet et al., 2019b). For example, if the two-dimensional prior $\theta = [\theta_1, \theta_2]$ is known to primarily influence the first component of the truth $\beta = [\theta_1 + \epsilon, \beta_2]$, we can select $\Lambda = \mathrm{diag}(d_1, d_2)$ with $d_1 \ll d_2$. This imposes a weaker penalty on perturbations in the first direction, resulting in a weighted penalty term: $\min_\kappa \big((\beta_1 - \kappa\theta_1)/d_1 + (\beta_2 - \kappa\theta_2)/d_2\big)$, which prioritizes aligning $\beta_1$ with $\theta_1$, while $\beta_2$ is determined more flexibly based on the data. We call this sub-coefficient-vector transferring, or the ability to partially transfer prior knowledge. A similar corollary applies to Theorem 3.6, as stated in Corollary D.1.

Finally, we again draw the reader's attention to Table 1, which compares several transfer learning methods discussed in Section 1.1.2. Notably, our proposed KG-WDRO framework brings together a broad range of desirable capabilities within a single, unified approach to transfer learning.

## 4. Numerical Results

In this section, we present numerical simulations to validate the effectiveness of the proposed KG-WDRO method. We compare learners across different settings, including high-dimensional sparse models, correlated covariates, and multi-source prior knowledge, for either linear regression or binary classification tasks. Performance is evaluated using out-of-sample classification error for binary classifiers and out-of-sample $R^2$ for linear regressors.

For the single-source experiments, target-source coefficient pairs $(\beta, \theta)$ are generated from a multivariate normal distribution:

$$(\beta_j, \theta_j) \sim N\left(\begin{pmatrix} 0 \\ 0 \end{pmatrix}, \begin{pmatrix} \sigma^2 & \rho\sigma^2 \\ \rho\sigma^2 & \sigma^2 \end{pmatrix}\right), \qquad (3)$$

where $\rho$ is the correlation between $\beta$ and $\theta$, and the expected length of $\theta$ is approximately $\sigma\sqrt{d - 0.5}$. We scale $\beta$ as $\beta \leftarrow s\beta$ with $s \in (0, 1]$ to study the stabilizing effects of strong prior knowledge in small-sample settings. The dimension-to-sample ratio $d/N$ is varied by fixing $d$ and increasing $N$. Performance is averaged over 100 simulations.

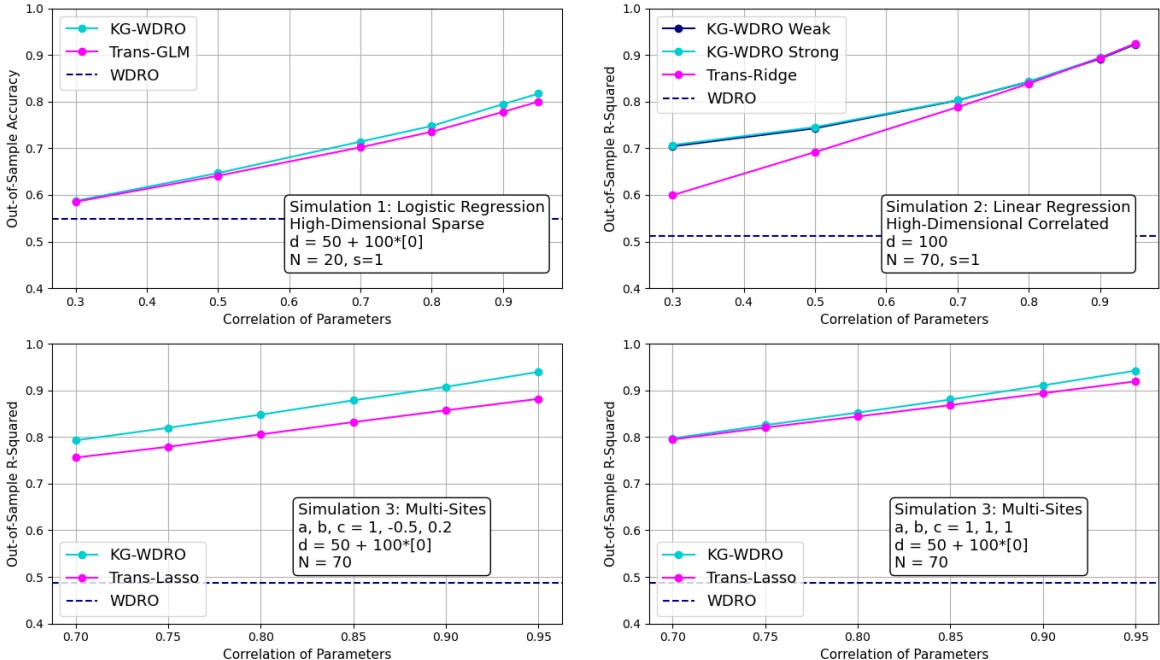

*Figure 2.* Out-of-sample performance plot of the proposed KG-WDRO method for high-dimensional regression tasks, compared against benchmark methods. The plot shows performance variations as $\rho$, representing the correlation between true and prior coefficient pairs, increases. Results are displayed for four specific settings across three experimental groups.

Each dataset consists of three parts: `data = (train, val, test)`. The `(train, val)` pair shares the same size, and hyperparameters are selected based on validation performance. The source data contains 800 samples, with source truth $\theta$ estimated accordingly. Out-of-sample performance is measured on the `test` set of 5000 data points.

### 4.1. Simulation 1: Logistic with $\ell_1$-Strong Transferring

In the first experiment, we compare two learners for binary classification tasks with high-dimensional sparse coefficients against our proposed KG-WDRO learner, $\beta_{\mathrm{KG}}$, derived using Theorem 3.6 with $p = 1$. The competing learners are the target-only vanilla WDRO learner $\beta_{\mathrm{WDRO}}$ (Blanchet et al., 2019a, Theorem 2) and $\beta_{\mathrm{TransG}}$, obtained via the $\mathcal{A}$-Trans-GLM algorithm (Tian & Feng, 2023, Algorithm 1). The target-source pair $(\beta, \theta)$ is generated using (3) with a dimension of 50 and augmented with 100 zeros for sparsity, resulting in a total dimension of 150. We test six settings, varying the sample size $N \in \{20, 50, 80\}$, signal strength $s \in \{0.5, 1\}$, and truth-prior correlation $\rho \in \{0.3, 0.5, 0.7, 0.8, 0.9, 0.95\}$.

The comparison between $\beta_{\mathrm{KG}}$ and $\beta_{\mathrm{TransG}}$ is highly competitive, with $\beta_{\mathrm{KG}}$ consistently outperforming $\beta_{\mathrm{TransG}}$ by up to 2% in accuracy when the sample size is small ($N = 20$) across all values of $\rho$, as shown in the upper-left plot of Figure 2. In larger sample size scenarios, both learners

perform similarly (see Table 3 for detailed results). Both transfer learning methods, $\beta_{\mathrm{KG}}$ and $\beta_{\mathrm{TransG}}$, significantly outperform the target-only learner, $\beta_{\mathrm{WDRO}}$.

### 4.2. Simulation 2: Linear Regression with $\ell_2$-Weak Transferring

In this simulation, we compare two learners on high-dimensional linear regression with correlated covariates against our proposed learners, $\beta_{\mathrm{KGweak}}$ (Theorem 3.5) and $\beta_{\mathrm{KGstrong}}$ (Theorem 3.3), both using $p = 2$. There is no sparsity in the regression coefficients. The competing learners are the target-only vanilla WDRO learner $\beta_{\mathrm{WDRO}}$ (Blanchet et al., 2019a, Theorem 1) and the Trans-Ridge algorithm adapted from (Li et al., 2021, Algorithm 1), denoted as $\beta_{\mathrm{TransR}}$. The covariates are fixed at dimension 100, with a pairwise correlation of 0.3. The experiment is conducted across six settings, varying the sample size $N \in \{50, 70, 90\}$, signal strength $s \in \{0.8, 1\}$, and truth-prior correlation $\rho \in \{0.3, 0.5, 0.7, 0.8, 0.9, 0.95\}$.

As shown in the upper-right plot of Figure 2, the performance of $\beta_{\mathrm{TransR}}$ lags significantly behind both $\beta_{\mathrm{KGstrong}}$ and $\beta_{\mathrm{KGweak}}$ until the correlation $\rho$ becomes sufficiently high. Across all settings, $\beta_{\mathrm{KGstrong}}$ and $\beta_{\mathrm{KGweak}}$ consistently outperform $\beta_{\mathrm{TransR}}$ when $\rho$ is moderate or low, as documented in Table 4. Furthermore, all three transfer learning methods demonstrate superior performance compared

### 4.3. Simulation 3: Transfer Learning with Multiple Sites

In the final set of experiments, we validate our methods in a multi-source transfer learning setting with high-dimensional sparse linear regression. The significant components of the three source coefficients are generated using (3) with correlation $\varrho$ and dimension 50, denoted as $\{\theta_1, \theta_2, \theta_3\}$. We construct a linear combination, $\theta_S = a\theta_1 + b\theta_2 + c\theta_3$, and generate $\beta = \rho\theta_S + \varepsilon$, where $\varepsilon \sim N(0, (1-\rho^2)\mathrm{Var}(\theta_S))$, ensuring $\mathrm{Corr}(\beta, \theta_S) = \rho$. $\beta$ is then scaled to match the magnitude of $\theta_S$, and all vectors are augmented with 100 zeros, yielding a total dimension of 150. Our proposed method, $\beta_{\mathrm{KG}}$ (Theorem 3.3, $p = 1$), is compared against the oracle Trans-Lasso algorithm (Li et al., 2021, Algorithm 1) ($\beta_{\mathrm{TransL}}$) and the vanilla WDRO learner $\beta_{\mathrm{WDRO}}$. The experiment spans six settings: $[a, b, c] = [1, -0.5, 0.2]$ and $[1, 1, 1]$, with $\varrho = 0.9$ and $0.6$, respectively. Sample sizes vary in $N \in \{50, 60, 70\}$. The truth-prior correlation ranges in $\rho \in \{0.7, 0.75, 0.8, 0.85, 0.9, 0.95\}$.

When $[a, b, c] = [1, -0.5, 0.2]$, the contributions of the $\theta$'s to the generation of $\theta_S$ are unequal. In this case, it is not surprising that $\beta_{\mathrm{KG}}$ outperforms $\beta_{\mathrm{TransL}}$, as shown in the bottom-left plot of Figure 2. When $\theta_S$ is an equal-weighted average of the $\theta$'s ($[a, b, c] = [1, 1, 1]$), the performance of $\beta_{\mathrm{KG}}$ and $\beta_{\mathrm{TransL}}$ becomes similar. However, $\beta_{\mathrm{KG}}$ still demonstrates superior performance in larger sample sizes and higher correlations, as documented in Table 5.

*Table 2.* Log-loss values for WDRO, KG-WDRO, and Trans-GLM across the eight target states and overall in the U.S. election dataset. The best-performing method in each state and overall is highlighted in distinct colors.

| STATE | WDRO | KG-WDRO | TRANS-GLM |
|---|---|---|---|
| ARIZONA | 22.43 | 7.54 | 8.52 |
| GEORGIA | 46.60 | 27.22 | 24.89 |
| ILLINOIS | 20.02 | 8.78 | 15.49 |
| MICHIGAN | 43.43 | 25.23 | 24.79 |
| MINNESOTA | 38.67 | 23.42 | 27.63 |
| MISSISSIPPI | 34.31 | 14.99 | 16.75 |
| N. CAROLINA | 41.48 | 19.02 | 18.56 |
| VIRGINIA | 64.84 | 21.20 | 22.84 |
| OVERALL | 311.77 | 147.39 | 159.49 |

### 4.4. A Real Data Analysis

To demonstrate the practical applicability of our KG-WDRO framework, we evaluate it on the Trans-GLM (Tian & Feng, 2023) dataset, which compiles 2020 U.S. presidential election results at the county level (see their references for data sources). Each county is labeled as '1' if the Democratic candidate won, and '0' otherwise. We compare KG-WDRO with Trans-GLM on a binary classification task, where the goal is to predict county-level election outcomes in eight target states (Table. 2) using data from the remaining states as external source knowledge. The features include county-level demographics such as population size and ethnicity proportions, and the base model is logistic regression. The cleaned dataset consists of 3,111 counties and 761 standardized predictors across 49 states. We use data from 2,100 counties as the source to predict outcomes in the eight target states (approximately 100 counties each). KG-WDRO outperforms Trans-GLM in 5 out of 8 states and reduces the overall log-loss by 7.6%. Both transfer learning methods significantly outperform the standard WDRO estimator.

## 5. Conclusion

We propose the knowledge-guided Wasserstein distributionally robust optimization (KG-WDRO) framework, which utilizes prior knowledge of predictors to mitigate the over-conservativeness of conventional DRO methods. We establish tractable reformulations and demonstrate their superior performance compared to other methods. For future work, we aim to provide statistical guarantees of our proposed estimators. Furthermore, based on these statistical properties, we plan to develop a principled approach for selecting hyperparameters such as $\delta$ and $\lambda$.

## Impact Statement

This paper presents work whose goal is to advance the field of Machine Learning. There might be potential societal consequences of our work, none of which we feel need to be specifically highlighted here.

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

# A. Additional Details in Numerical Results

This section provides details to supplement Section 4. We outline the data-generating distributions for all three sets of experiments, the hyperparameter grids, and the learners used to identify prior knowledge. We present the exact numerical results for all three sets of experiments. Recall that the notation $s \in (0, 1]$ represents the signal strength of the true parameter $\beta$, which works by rescaling the magnitude of $\beta$ such that $\beta \leftarrow s\beta$. The notation $d$ is the dimension of the covariates, and $N$ is the sample size. Finally, the symbol $\rho$ represents the correlation between the true $\beta$ and the prior $\theta$.

## A.1. Simulation Results

### A.1.1. SIMULATION 1: LOGISTIC REGRESSION

| SETTING | | $\rho = 0.3$ | $\rho = 0.5$ | $\rho = 0.7$ | $\rho = 0.8$ | $\rho = 0.9$ | $\rho = 0.95$ | WDRO |
|---|---|---|---|---|---|---|---|---|
| $s = 1$ | KG-WDRO | **0.587** | **0.647** | **0.714** | **0.748** | **0.794** | **0.817** | 0.565 |
| $N = 20$ | TRANS-GLM | 0.585 | 0.641 | 0.702 | 0.735 | 0.778 | 0.800 | - |
| $s = 1$ | KG-WDRO | 0.586 | **0.647** | 0.713 | 0.751 | **0.797** | 0.823 | 0.619 |
| $N = 50$ | TRANS-GLM | **0.586** | 0.645 | 0.710 | **0.752** | 0.792 | **0.823** | - |
| $s = 1$ | KG-WDRO | 0.583 | **0.646** | 0.713 | 0.751 | 0.798 | 0.823 | 0.654 |
| $N = 80$ | TRANS-GLM | **0.584** | 0.646 | **0.714** | **0.755** | **0.800** | **0.826** | - |
| $s = 0.5$ | KG-WDRO | **0.581** | **0.634** | **0.690** | **0.721** | **0.762** | **0.787** | 0.549 |
| $N = 20$ | TRANS-GLM | 0.579 | 0.626 | 0.674 | 0.708 | 0.748 | 0.760 | - |
| $s = 0.5$ | KG-WDRO | **0.580** | **0.635** | 0.689 | **0.728** | 0.768 | **0.794** | 0.588 |
| $N = 50$ | TRANS-GLM | 0.579 | 0.633 | **0.693** | 0.723 | 0.769 | 0.789 | - |
| $s = 0.5$ | KG-WDRO | 0.581 | 0.637 | 0.700 | 0.732 | 0.775 | 0.790 | 0.617 |
| $N = 80$ | TRANS-GLM | 0.581 | **0.638** | **0.702** | **0.737** | **0.779** | **0.799** | - |

*Table 3.* Out-of-sample classification accuracies for Simulation 4.1, comparing KG-WDRO, Trans-GLM, and WDRO across six settings with varying values of $\rho$.

### A.1.2. SIMULATION 2: LINEAR REGRESSION

| SETTING | | $\rho = 0.3$ | $\rho = 0.5$ | $\rho = 0.7$ | $\rho = 0.8$ | $\rho = 0.9$ | $\rho = 0.95$ | WDRO |
|---|---|---|---|---|---|---|---|---|
| $s = 1$ | KG-WDRO (WEAK) | **0.585** | 0.645 | 0.740 | **0.801** | 0.870 | 0.912 | 0.108 |
| $N = 50$ | KG-WDRO (STRONG) | 0.583 | **0.646** | **0.741** | 0.800 | **0.871** | 0.910 | - |
| | TRANS-RIDGE | 0.391 | 0.548 | 0.706 | 0.786 | 0.870 | **0.915** | - |
| $s = 1$ | KG-WDRO (WEAK) | **0.707** | **0.745** | **0.803** | **0.843** | **0.894** | 0.924 | 0.513 |
| $N = 70$ | KG-WDRO (STRONG) | 0.704 | 0.743 | 0.803 | 0.842 | 0.892 | 0.923 | - |
| | TRANS-RIDGE | 0.599 | 0.692 | 0.788 | 0.838 | 0.893 | **0.925** | - |
| $s = 1$ | KG-WDRO (WEAK) | **0.806** | **0.827** | **0.859** | **0.881** | **0.911** | **0.932** | 0.758 |
| $N = 90$ | KG-WDRO (STRONG) | 0.804 | 0.825 | 0.857 | 0.880 | 0.910 | 0.930 | - |
| | TRANS-RIDGE | 0.762 | 0.802 | 0.849 | 0.877 | 0.910 | **0.932** | - |
| $s = 0.8$ | KG-WDRO (WEAK) | **0.563** | 0.621 | **0.716** | **0.777** | **0.850** | **0.894** | 0.030 |
| $N = 50$ | KG-WDRO (STRONG) | 0.561 | **0.622** | 0.716 | 0.777 | 0.849 | 0.892 | - |
| | TRANS-RIDGE | 0.213 | 0.405 | 0.600 | 0.700 | 0.803 | 0.858 | - |
| $s = 0.8$ | KG-WDRO (WEAK) | **0.673** | **0.713** | **0.774** | **0.818** | **0.872** | **0.905** | 0.361 |
| $N = 70$ | KG-WDRO (STRONG) | 0.670 | 0.710 | 0.774 | 0.816 | 0.869 | 0.903 | - |
| | TRANS-RIDGE | 0.470 | 0.585 | 0.704 | 0.768 | 0.837 | 0.875 | - |
| $s = 0.8$ | KG-WDRO (WEAK) | **0.768** | **0.791** | **0.826** | **0.851** | **0.886** | **0.911** | 0.703 |
| $N = 90$ | KG-WDRO (STRONG) | 0.765 | 0.788 | 0.825 | 0.851 | 0.885 | 0.909 | - |
| | TRANS-RIDGE | 0.671 | 0.724 | 0.785 | 0.821 | 0.863 | 0.890 | - |

*Table 4.* Out-of-sample $R^2$ for Simulation 4.2, comparing KG-WDRO (Strong), KG-WDRO (Weak), Trans-Ridge, and WDRO across six settings with varying values of $\rho$.

### A.1.3. SIMULATION 3: MULTI-SITES

Here, recall that the notation $\varrho$ denote the correlation of generating the three prior knowledge under the scheme (3).

| SETTING | | $\rho = 0.7$ | $\rho = 0.75$ | $\rho = 0.8$ | $\rho = 0.85$ | $\rho = 0.9$ | $\rho = 0.95$ | WDRO |
|---|---|---|---|---|---|---|---|---|
| $[1, -0.5, 0.2]$ | KG-WDRO | 0.560 | **0.640** | **0.713** | **0.783** | **0.850** | **0.916** | -0.584 |
| $\varrho = 0.9, N = 50$ | TRANS-LASSO | **0.578** | 0.625 | 0.673 | 0.723 | 0.767 | 0.815 | - |
| $[1, -0.5, 0.2]$ | KG-WDRO | **0.674** | **0.728** | **0.776** | **0.825** | **0.875** | **0.926** | 0.027 |
| $\varrho = 0.9, N = 60$ | TRANS-LASSO | 0.666 | 0.697 | 0.732 | 0.770 | 0.808 | 0.850 | - |
| $[1, -0.5, 0.2]$ | KG-WDRO | **0.793** | **0.820** | **0.848** | **0.878** | **0.907** | **0.939** | 0.375 |
| $\varrho = 0.9, N = 70$ | TRANS-LASSO | 0.756 | 0.779 | 0.805 | 0.832 | 0.857 | 0.882 | - |
| $[1, 1, 1]$ | KG-WDRO | 0.565 | 0.642 | 0.715 | 0.785 | **0.852** | **0.916** | -2.837 |
| $\varrho = 0.6, N = 50$ | TRANS-LASSO | **0.628** | **0.680** | **0.735** | **0.790** | 0.838 | 0.889 | - |
| $[1, 1, 1]$ | KG-WDRO | 0.673 | 0.729 | 0.778 | **0.829** | **0.877** | **0.928** | -0.015 |
| $\varrho = 0.6, N = 60$ | TRANS-LASSO | **0.708** | **0.744** | **0.786** | 0.826 | 0.863 | 0.902 | - |
| $[1, 1, 1]$ | KG-WDRO | **0.797** | **0.825** | **0.852** | **0.880** | **0.911** | **0.942** | 0.354 |
| $\varrho = 0.6, N = 70$ | TRANS-LASSO | 0.794 | 0.820 | 0.844 | 0.868 | 0.894 | 0.919 | - |

*Table 5.* Out-of-sample $R^2$ for Simulation 4.3, comparing KG-WDRO, Trans-Lasso, and WDRO across six settings with varying values of $\rho$.

## A.2. Simulation Setup

Let $\mathrm{Ber}(p)$ denote a bernoulli distribution with probability parameter $p$, $\mathcal{U}[a, b]$ denote a uniform distribution supported on $[a, b]$, and $\mathcal{N}(\mu, \sigma^2)$ denote a univariate normal distribution with mean $\mu$ and variance $\sigma^2$.

### A.2.1. SIMULATION 1: LOGISTIC REGRESSION

In this simulation, the coefficients are generated in a high-dimensional sparse setting. The dimension of the nonzero components is set to 50, which is then augmented with 100 zero components to introduce sparsity. The nonzero components of the true coefficient-prior pair $(\beta, \theta)$ are generated using the multivariate normal scheme in (3), with component variance $\sigma^2 = 0.4$ and $\rho \in \{0.3, 0.5, 0.7, 0.8, 0.9, 0.95\}$. The target labels are generated as $Y_{\mathrm{target}} \sim \mathrm{Ber}\left(1/(1 + \exp\left(-\beta^{\mathsf{T}} X\right)\right)$, and the source labels are generated as $Y_{\mathrm{source}} \sim \mathrm{Ber}\left(1/(1 + \exp\left(-\theta^{\mathsf{T}} X\right)\right)$, where $X \sim \mathcal{U}[-2, 2]^{150}$. The sample size $N$ for $(X_{\mathrm{target}}, Y_{\mathrm{target}})$ is varied across $\{20, 50, 80\}$, while the sample size for the source data $(X_{\mathrm{source}}, Y_{\mathrm{source}})$ is fixed at 800. Each dataset is paired with a validation set of the same size for hyperparameter selection.

Let $\mathrm{grid}_1$ denote a hyperparameter grid ranging from 0.0001 to 1 with 10 log-spaced values, and let $\mathrm{grid}_2$ denote a hyperparameter grid ranging from 0.0001 to 2 with 20 log-spaced values. The $\beta_{\mathrm{WDRO}}$ estimator is learned by selecting the best-performing hyperparameter on $\mathrm{grid}_1$ using validation data. For the $\mathcal{A}$-Trans-GLM learner (Tian & Feng, 2023, Algorithm 1), the transferring step is optimized using $\mathrm{grid}_1$, and the debiasing step is optimized using $\mathrm{grid}_2$. For the KG-WDRO learner $\beta_{\mathrm{KG}}$ proposed in Theorem 3.6 with $p = 1$, the prior $\theta$ is first learned from the source data using the vanilla WDRO method on $\mathrm{grid}_1$, followed by learning $\beta_{\mathrm{KG}}$ on $\mathrm{grid}_2$ with the learned $\theta_{\mathrm{WDRO}}$ as input.

The simulations are conducted on the parameter grid $N \in \{20, 50, 80\} \times \rho \in \{0.3, 0.5, 0.7, 0.8, 0.9, 0.95\} \times s \in \{0.5, 1\}$, with each configuration repeated 100 times. The average results are reported.

### A.2.2. SIMULATION 2: LINEAR REGRESSION

In this simulation, the coefficients are generated in a high-dimensional correlated setting. The dimension of the coefficients is set to 100 and the components of the true coefficient-prior pair $(\beta, \theta)$ are generated using the multivariate normal scheme in (3), with component variance $\sigma^2 = 0.1$ and $\rho \in \{0.3, 0.5, 0.7, 0.8, 0.9, 0.95\}$. The target labels are generated as $Y_{\mathrm{target}} \sim \mathcal{N}(\beta^{\mathsf{T}} X, \sqrt{0.5})$, and the source labels are generated as $Y_{\mathrm{source}} \sim \mathcal{N}(\theta^{\mathsf{T}} X, \sqrt{0.5})$, where $X \sim \mathcal{N}(\mathbf{0}, \Sigma)$ with

$$\Sigma_{i,j} = \begin{cases} 1 & \text{if } i = j, \\ 0.3 & \text{if } i \neq j, \end{cases} \quad \text{for all } i, j = 1, 2, \ldots, 100.$$

The sample size $N$ for $(X_{\text{target}}, Y_{\text{target}})$ is varied across $\{50, 70, 90\}$, while the sample size for the source data $(X_{\text{source}}, Y_{\text{source}})$ is fixed at 800. Each dataset is paired with a validation set of the same size for hyperparameter selection.

Let $\text{grid}_1$ denote a hyperparameter grid ranging from 0.0001 to 1 with 10 log-spaced values, and let $\text{grid}_2$ denote a hyperparameter grid ranging from 0.0001 to 1.5 with 20 log-spaced values. The $\beta_{\text{WDRO}}$ estimator is learned by selecting the best-performing hyperparameter on $\text{grid}_1$ using validation data. For the Trans-Ridge learner adapted from (Li et al., 2021, Algorithm 1), the transferring step is optimized using $\text{grid}_1$, and the debiasing step is optimized using $\text{grid}_2$. For the KG-WDRO learner $\beta_{\text{KGstrong}}$ proposed in Theorem 3.3 with $p = 2$, and the $\beta_{\text{KGweak}}$ learner proposed in Theorem 3.5, the prior $\theta$ is first learned from the source data using the vanilla WDRO method on $\text{grid}_1$, followed by learning $\beta_{\text{KGstrong}}$ on $\text{grid}_2$ with the learned $\theta_{\text{WDRO}}$ as input. The $\lambda^{-1}$ grid for $\beta_{\text{KGweak}}$ is 0.0001 to 8 with 20 log-spaced values.

The simulations are conducted on the parameter grid $N \in \{50, 70, 90\} \times \rho \in \{0.3, 0.5, 0.7, 0.8, 0.9, 0.95\} \times s \in \{0.8, 1\}$, with each configuration repeated 100 times. The average results are reported.

### A.2.3. SIMULATION 3: MULTIPLE SITES

In this simulation, the coefficients are generated in a high-dimensional sparse setting. The dimension of the nonzero components is set to 50, which is then augmented with 100 zero components to introduce sparsity. The number of external source is 3, we generate the their coefficients $\theta_1, \theta_2, \theta_3$ using the scheme (3). We construct a linear combination, $\theta_S = a\theta_1 + b\theta_2 + c\theta_3$, and generate the target coefficient $\beta = \rho\theta_S + \varepsilon$, where $\varepsilon \sim N(0, (1 - \rho^2)\text{Var}(\theta_S))$, ensuring $\text{Corr}(\beta, \theta_S) = \rho$. The target coefficient $\beta$ is then scaled to match the magnitude of $\theta_S$.

The target labels are generated as $Y_{\text{target}} \sim \mathcal{N}(\beta^\intercal X, \sqrt{0.5})$, and the source labels are generated as $Y_{\text{source}, m} \sim \mathcal{N}(\theta_m^\intercal X, \sqrt{0.5})$ for $m \in [3]$, where $X \sim \mathcal{N}(\mathbf{0}, \Sigma)$ with

$$\Sigma_{i,j} = \begin{cases} 1 & \text{if } i = j, \\ 0.1 & \text{if } i \neq j, \end{cases} \quad \text{for all } i, j = 1, 2, \ldots, 150.$$

The sample size for the target data ranges in $\{50, 60, 70\}$.

Let $\text{grid}_1$ denote a hyperparameter grid ranging from 0.0001 to 1 with 15 log-spaced values, and let $\text{grid}_2$ denote a hyperparameter grid ranging from 0.0001 to 3 with 20 log-spaced values. The $\beta_{\text{WDRO}}$ estimator is learned by selecting the best-performing hyperparameter on $\text{grid}_1$ using validation data. For the oracle Trans-Lasso learner (Li et al., 2021, Algorithm 1), the transferring step is optimized using $\text{grid}_1$, and the debiasing step is optimized using $\text{grid}_2$ using all three source data. For the KG-WDRO learner $\beta_{\text{KG}}$ proposed in Theorem 3.3 with $p = 1$, the priors $\theta_1, \theta_2, \theta_3$ are first learned from the three source data using the vanilla WDRO method on $\text{grid}_1$, followed by learning $\beta_{\text{KG}}$ on $\text{grid}_2$ with the learned $\theta_{1,\text{WDRO}}, \theta_{2,\text{WDRO}}, \theta_{3,\text{WDRO}}$ as input.

The simulations are conducted on the parameter grid $N \in \{50, 60, 70\} \times \rho \in \{0.3, 0.5, 0.7, 0.8, 0.9, 0.95\} \times [a, b, c] \in \{[1, -0.5, 0.2], [1, 1, 1]\}$, with each configuration repeated 100 times. The average results are reported.

## B. Proof of Results in Regression.

***Proof of Proposition 3.1.*** This result follows from the observation that, under the constraint imposed by $c_{2,\infty}$, for any $\mathbb{P}^\# \in \mathcal{B}_\delta(\mathbb{P}_N; c_{2,\infty})$ and any $\alpha \in \mathbb{R}$, the marginal distributions of $(Y, \alpha\theta^\intercal X)$ must agree under $\mathbb{P}_N$ and $\mathbb{P}^\#$. That is,

$$\mathbb{P}_N\big((Y, \alpha\theta^\intercal X) \in A \times B\big) = \mathbb{P}^\#\big((Y, \alpha\theta^\intercal X) \in A \times B\big),$$

for all Borel measurable sets $A, B \subset \mathbb{R}$. Consequently, for any $\mathbb{P}^\# \in \mathcal{B}_\delta(\mathbb{P}_N; c_{2,\infty})$ we have

$$\mathbb{E}_{\mathbb{P}_N}\big[(Y - \alpha\theta^\intercal X)^2\big] = \mathbb{E}_{\mathbb{P}^\#}\big[(Y - \alpha\theta^\intercal X)^2\big],$$

then choosing $\beta = \alpha\theta$, we have

$$\begin{aligned}
\inf_{\beta \in \mathbb{R}^d} \sup_{\mathbb{P}: \mathcal{B}_\delta(\mathbb{P}_N)} \mathbb{E}_{\mathbb{P}}\big[(Y - \beta^\intercal X)^2\big] &= \sup_{\mathbb{P}: \mathcal{B}_\delta(\mathbb{P}_N)} \inf_{\beta \in \mathbb{R}^d} \mathbb{E}_{\mathbb{P}}\big[(Y - \beta^\intercal X)^2\big] \\
&\leq \sup_{\mathbb{P}: \mathcal{B}_\delta(\mathbb{P}_N)} \mathbb{E}_{\mathbb{P}}\big[(Y - \alpha\theta^\intercal X)^2\big]. \\
&= \mathbb{E}_{\mathbb{P}_N}\big[(Y - \alpha\theta^\intercal X)^2\big],
\end{aligned}$$

where the first equality invoked the Minimax Theorem of WDRO (Blanchet et al., 2019a, Lemma 1). Taking infimum over $\alpha$ completes the proof. □

**Lemma B.1.** *Let $f_\beta : \mathbb{R}^d \to \mathbb{R}$ be defined as $\Delta \in \mathbb{R}^d \mapsto (\beta^\mathsf{T}\Delta)^2 - 2r(\beta)\beta^\mathsf{T}\Delta$ depending on some $\beta \in \mathbb{R}^d$ and let $r(\beta)$ be a non-negative real-valued function in $\beta$. Then the convex conjugate $f_\beta^*(\Delta^*) : \mathbb{R}^d \to \mathbb{R}$ is given by*

$$f_\beta^*(\Delta^*) = \begin{cases} \dfrac{(\beta^\mathsf{T}\Delta^* + 2r(\beta)\|\beta\|_2^2)^2}{4\|\beta\|_2^4} & \text{if } \Delta^* \in \operatorname{span}\beta, \\ +\infty & \text{otherwise.} \end{cases}$$

*Therefore the biconjugate $f_\beta^{**}(\Delta) : \mathbb{R}^d \to \mathbb{R}$ of $f_\beta(\Delta)$ has representation:*

$$f_\beta^{**}(\Delta) = \sup_{\alpha \in \mathbb{R}} \left( \alpha(\beta^\mathsf{T}\Delta) - \frac{(\alpha + 2r(\beta))^2}{4} \right).$$

*Proof.* The convex conjugate $f_\beta^*(\Delta^*)$ is defined as

$$f_\beta^*(\Delta^*) := \sup_{\Delta \in \mathbb{R}^d} \left( \Delta^{*\mathsf{T}}\Delta - (\beta^\mathsf{T}\Delta)^2 + 2r(\beta)(\beta^\mathsf{T}\Delta) \right),$$

where $\Delta^*, \beta \in \mathbb{R}^d$ and $r(\beta) \in \mathbb{R}$ are taken as fixed values. Orthogonalize $\Delta = a\beta + \omega$ in the direction of $\beta$ with $a \in \mathbb{R}$, and $\omega \in \mathbb{R}^d$ such that $\beta^\mathsf{T}\omega = 0$. Then , we have $\Delta^{*\mathsf{T}}\Delta = a\Delta^{*\mathsf{T}}\beta + \Delta^{*\mathsf{T}}\omega$, and the convex conjugate becomes

$$f^*(\Delta^*) = \sup_{a,\omega} \left( a(\Delta^{*\mathsf{T}}\beta) + \Delta^{*\mathsf{T}}\omega - a^2\|\beta\|_2^4 + 2ar(\beta)\|\beta\|_2^2 \right)$$
$$\text{s.t} \quad \beta^\mathsf{T}\omega = 0.$$

Fixing $\omega$, the objective is a negative quadratic function in $a$, hence the objective in $a$ is bounded from above by a finite value. Now, if $\Delta^*$ is not orthogonal to $\omega$, the term $\sup_\omega \Delta^{*\mathsf{T}}\omega$ is unbounded, and the convex conjugate $f^*(\Delta^*) = +\infty$. If $\Delta^*$ is orthogonal to $\omega$, then the convex conjugate attains finite value. Note that $\Delta^{*\mathsf{T}}\omega = 0 \iff \Delta^* \in \operatorname{span}\beta$. Hence condition on $\{\Delta^* = \alpha\beta \,;\, \alpha \in \mathbb{R}\}$, we have

$$f^*(\Delta^*) = \sup_a \left( a(\Delta^{*\mathsf{T}}\beta) - a^2\|\beta\|_2^4 + 2r(\beta)a\|\beta\|_2^2 \right)$$
$$= \frac{\left( \Delta^{*\mathsf{T}}\beta + 2r(\beta)\|\beta\|_2^2 \right)^2}{4\|\beta\|_2^4},$$

where the optimal solution is $a^* = \dfrac{\alpha + 2r(\beta)}{2\|\beta\|_2^2}$, and the coefficient $\alpha$ is given by the projection scalar $\alpha = \dfrac{\Delta^{*\mathsf{T}}\beta}{\|\beta\|_2^2}$.

The biconjuagte

$$f^{**}(\Delta) = \sup_{\Delta^*} \left( \Delta^\mathsf{T}\Delta^* - f^*(\Delta^*) \right),$$

is therefore bounded from below if and only if $\Delta^* \in \operatorname{span}\beta$. Let $\Delta^* = \alpha\beta$ for some $\alpha \in \mathbb{R}$, then substituting we get the representation,

$$f^{**}(\Delta) = \sup_\alpha \left( \Delta^\mathsf{T}(\alpha\beta) - \frac{\left( \beta^\mathsf{T}(\alpha\beta) + 2r(\beta)\|\beta\|_2^2 \right)^2}{4\|\beta\|_2^4} \right)$$
$$= \sup_\alpha \left( \alpha(\Delta^\mathsf{T}\beta) - \frac{(\alpha + 2r(\beta))^2}{4} \right).$$

It can be readily verified that $f^{**}(\Delta) = f(\Delta)$ as promised by the *Fenchel-Moreau Theorem* (Theorem E.4). □

**Lemma B.2.** *Let $g_\theta(\Delta) : \mathbb{R}^d \to \mathbb{R}$ be defined as $\Delta \in \mathbb{R}^d \mapsto |\theta^\mathsf{T}\Delta|$ for some $\theta \in \mathbb{R}^d$. Then the convex conjugate $g_\theta^*(\Delta^*)$ is given by*

$$g_\theta^*(\Delta^*) = \begin{cases} 0 & \text{if } \Delta^* = \alpha\theta \text{ and } |\alpha| \leq 1, \\ +\infty & \text{otherwise.} \end{cases}$$

*Therefore the convex conjugate of the function $g(\Delta) := \gamma \sum_{m=1}^{M} g_{\theta_m}(\Delta)$ for some $\gamma > 0$ is given by*

$$g^*(\Delta^*) = \begin{cases} 0 & \text{if } \Delta^* = \sum_{m=1}^{M} \alpha_m \theta_m \text{ and }, |\alpha_m| \leq \gamma \text{ for each } m, \\ +\infty & \text{otherwise.} \end{cases}$$

*Proof.* The convex conjugate is defined as

$$g_\theta^*(\Delta^*) = \sup_{\Delta} \left( \Delta^{*\mathsf{T}}\Delta - |\theta^\mathsf{T}\Delta| \right),$$

again, orthogonalize $\Delta = a\theta + \omega$, where $a = \dfrac{\theta^\mathsf{T}\Delta}{\|\theta\|_2^2}$ and $\theta^\mathsf{T}\omega = 0$. Now by the change of variable $u := \theta^\mathsf{T}\Delta$, the convex conjugate is now

$$g_\theta^*(\Delta^*) = \sup_{u,\omega} \left( \frac{u}{\|\theta\|_2^2}(\Delta^{*\mathsf{T}}\theta) + \Delta^{*\mathsf{T}}\omega - |u| \right)$$
$$\text{s.t} \quad \theta^\mathsf{T}\omega = 0.$$

Thus the convex conjugate $g_\theta^*(\Delta^*) = +\infty$ if $\Delta^* \notin \operatorname{span}\theta$. If $\Delta^* = \alpha\theta$ for some $\alpha \in \mathbb{R}$, then

$$g_\theta^*(\Delta^*) = g_\theta^*(\alpha\theta) = \sup_u \left( \frac{u}{\|\theta\|_2^2}\alpha\|\theta\|_2^2 - |u| \right)$$
$$= \sup_u \left( \alpha u - |u| \right)$$
$$= \begin{cases} 0 & \text{if } |\alpha| \leq 1, \\ +\infty & \text{otherwise,} \end{cases}$$

where the last equality holds by noting that $\sup_u \alpha u - |u| = |\cdot|^*(\alpha)$ is the convex conjugate of the absolute value function (Proposition E.2). This proofs the convex conjugate of $g_\theta^*(\Delta^*)$. Now $g(\Delta) = \gamma \sum_{m=1}^{M} g_{\theta_m}(\Delta) = \gamma\bar{g}(\Delta)$, the convex conjugate of $\bar{g}(\Delta)$ is

$$\bar{g}^*(\Delta^*) = (g_{\theta_1} + \ldots + g_{\theta_M})^*(\Delta^*)$$
$$= \inf_{\Delta^*} \left( g_{\theta_1}^*(\Delta_1^*) + \ldots + g_{\theta_M}^*(\Delta_M^*) \right) \quad \text{s.t } \Delta_1^* + \ldots + \Delta_M^* = \Delta^*,$$

where the second line follows from the *infimal convolution property* of sum of convex conjugates (Theorem E.5). We know that $\bar{g}^*$ is finite if and only if $g_{\theta_m}^*(\Delta_m^*) = 0$ for all $m \in [M]$, that is $\Delta_m^* = \alpha_m \theta_m$ for some $\alpha_m \in [-1, 1]$ for all $m \in [M]$. Hence $\bar{g}^*(\Delta^*) = 0$ if and only if $\Delta^* = \sum_{m=1}^{M} \alpha_m \theta_m$ and $\alpha_m \in [-1, 1]$ for all $m \in [M]$. Finally we can calculate the convex conjugate $g^*(\Delta^*) = (\gamma\bar{g})^*(\Delta^*) = \gamma\bar{g}^* \left( \dfrac{\Delta^*}{\gamma} \right)$ by the scaling law of convex conjugates (Proposition E.3) given $\gamma > 0$. This concludes the proof. $\qquad\square$

We now give the proof to Theorem 3.3.

***Proof of Theorem 3.3.*** Let $r(\beta) := y - \beta^\mathsf{T}x$. Then first consider the cost function

$$c_2\big((x,y),(u,v)\big) := \|x - u\|_q^2 + \infty \cdot |y - v| + d(\theta_1^\mathsf{T}x - \theta_1^\mathsf{T}u) + \ldots + d(\theta_M^\mathsf{T}x - \theta_M^\mathsf{T}u).$$

where we replaced the transferring strength from $+\infty$ to a finite-valued distance function $d(x) : \mathbb{R} \to \mathbb{R}$ that is a monotone function in $|x|$, with $d(0) = 0$. We will then let $d(x) \to \infty$ except at $x = 0$. Then the supremum function

$$\phi_\gamma(x, y; \beta) = \sup_{(u,v) \in \mathbb{R}^{d+1}} \left\{ \ell(u, v; \beta) - \gamma c\big((u,v),(x,y)\big) \right\},$$

is finite if and only if $v = y$. Then, we have

$$l(u, v; \beta) - \gamma c\left((u,v),(x,y)\right)$$
$$= (y - \beta^\mathsf{T}u)^2 - \gamma\|x - u\|_q^2 - \gamma d\left(\theta_1^\mathsf{T}x - \theta_1^\mathsf{T}u\right) - \ldots - \gamma d(\theta_M^\mathsf{T}x - \theta_M^\mathsf{T}u).$$

Denote by $\Delta := u - x$, we get

$$
\begin{aligned}
&l\left(u, v; \beta\right) - \gamma c\left((u, v), (x, y)\right) \\
&= r(\beta)^2 + \left\{(\beta^\mathsf{T}\Delta)^2 - 2r(\beta)\beta^\mathsf{T}\Delta - \gamma\|\Delta\|_q^2 - \gamma d(\theta_1^\mathsf{T}\Delta) - \ldots - \gamma d(\theta_M^\mathsf{T}\Delta)\right\}.
\end{aligned}
$$

Consider the objective in $\Delta$

$$
\begin{aligned}
&\sup_\Delta \left\{(\beta^\mathsf{T}\Delta)^2 - 2r(\beta)\beta^\mathsf{T}\Delta - \gamma\|\Delta\|_q^2 - \gamma d(\theta_1^\mathsf{T}\Delta) - \ldots - \gamma d(\theta_M^\mathsf{T}\Delta)\right\} \\
&:= \sup_\Delta \left\{f_\beta(\Delta) - g(\Delta)\right\},
\end{aligned}
$$

where we let $f_\beta(\Delta) := (\beta^\mathsf{T}\Delta)^2 - 2r(\beta)\beta^\mathsf{T}\Delta$ and $g(\Delta) := \gamma\|\Delta\|_q^2 + \gamma d(\theta_1^\mathsf{T}\Delta) + \ldots + \gamma d(\theta_M^\mathsf{T}\Delta)$. This is a convex + concave optimization, we express the convex part of $f_\beta(\Delta)$ as a supremum of infinitely many affine functions. Then by Lemma B.1, we have $f_\beta(\Delta) = f_\beta^{**}(\Delta) = \sup_{\alpha \in \mathbb{R}}\left(\alpha(\beta^\mathsf{T}\Delta) - \frac{(\alpha + 2r(\beta))^2}{4}\right)$, then we may write

$$
\begin{aligned}
&\sup_\Delta \left\{f_\beta(\Delta) - g(\Delta)\right\} \\
&= \sup_\Delta \left\{\sup_{\alpha \in \mathbb{R}}\left(\alpha(\beta^\mathsf{T}\Delta) - \frac{(\alpha + 2r(\beta))^2}{4}\right) - g(\Delta)\right\} \\
&= \sup_\alpha \left\{\sup_\Delta \left(\Delta^\mathsf{T}(\alpha\beta) - g(\Delta)\right) - \frac{(\alpha + 2r(\beta))^2}{4}\right\} \\
&= \sup_\alpha \left\{g^*(\alpha\beta) - \frac{(\alpha + 2r(\beta))^2}{4}\right\}, \qquad\qquad\qquad\text{(Toland's Duality)}
\end{aligned}
$$

where $g^*$ is the convex conjugate of $g$. Let $g(\Delta) := g_1(\Delta) + g_\theta(\Delta)$, with $g_1(\Delta) = \gamma\|\Delta\|_q^2$ and $g_\theta(\Delta) := \gamma \sum_{m=1}^{M} d(\theta_m^\mathsf{T}\Delta)$. Then we can compute the convex conjugate of $g$ using the *infimal convolution property* (Theorem E.5). Then

$$
g^*(\Delta^*) = \inf_{\Delta_1^* + \Delta_2^* = \Delta^*} \left(g_1^*(\Delta_1^*) + g_\theta^*(\Delta_2^*)\right).
$$

We know that $g_1^*(\Delta_1^*) = \frac{1}{4\gamma}\|\Delta_1^*\|_p^2$, where $p^{-1} + q^{-1} = 1$ (Proposition E.2). Now suppose $d(x) = \lambda|x|$ for some $\lambda > 0$, by Lemma B.2, we have,

$$
g_\theta^*(\Delta_2^*) = \begin{cases} 0 & \text{if } \Delta_2^* = \sum_{m=1}^{M} \alpha_m \theta_m \text{ and }, |\alpha_m| \leq \gamma\lambda \text{ for each } m, \\ +\infty & \text{otherwise.} \end{cases}
$$

Then the convex conjugate $g^*(\Delta^*)$ is

$$
g^*(\Delta^*) = \inf_{\Delta_2^*} g_1^*(\Delta^* - \Delta_2^*),
$$

$$
\text{s.t } \Delta_2^* = \sum_{m=1}^{M} \alpha_m \theta_m \text{ and }, |\alpha_m| \leq \gamma\lambda \text{ for each } m,
$$

which is equivalently,

$$
g^*(\Delta^*) = \frac{1}{4\gamma} \inf_{\boldsymbol{\alpha}} \left\|\Delta^* - \sum_{m=1}^{M} \alpha_m \theta_m\right\|_p^2,
$$

$$
\text{s.t } |\alpha_m| \leq \gamma\lambda \text{ for each } m.
$$

Letting $\lambda \to \infty$, we recover the cost function $c_{2,\infty}$, and when $\lambda \to \infty$, each $\alpha_m$ is now free in $\mathbb{R}$. Then we have $g^*(\Delta^*) = \frac{1}{4\gamma} \inf_{\vartheta \in \Theta} \|\Delta^* - \vartheta\|_p^2$, with $\Theta := \text{span}\{\theta_1, \ldots, \theta_M\}$, the validity of this tactic follows from (Luenberger & Ye,

2008, Theorem 1, Section 13.1). Then we have $g^*(\alpha\beta) = \frac{1}{4\gamma}\inf_{\vartheta\in\Theta}\|\alpha\beta - \vartheta\|_p^2$. Suppose $\alpha \neq 0$, then dividing by $\alpha$, we get

$$g^*(\alpha\beta) = \frac{\alpha^2}{4\gamma}\inf_{\vartheta\in\Theta}\|\beta - \vartheta\|_p^2.$$

If $\alpha = 0$, then $g^*(\alpha\beta) = g^*(\mathbf{0}) = \frac{1}{4\gamma}\inf_{\vartheta}\|\vartheta\|_p^2 = 0$, so the representation $g^*(\alpha\beta) = \frac{\alpha^2}{4\gamma}\inf_{\vartheta\in\Theta}\|\beta - \vartheta\|_p^2$, is valid for all $\alpha \in \mathbb{R}$. Therefore following the proof of (Blanchet et al., 2019a, Theorem 1),

$$\phi_\gamma(x,y;\beta) = r(\beta)^2 + \frac{1}{4}\sup_\alpha\left\{\frac{\alpha^2}{\gamma}\inf_{\vartheta\in\Theta}\|\beta - \vartheta\|_p^2 - \left(\alpha + 2r(\beta)\right)^2\right\}$$

$$= \frac{1}{4}\sup_\alpha\left\{\left(\frac{\inf_\vartheta\|\beta - \vartheta\|_p^2}{\gamma} - 1\right)\alpha^2 - 4r(\beta)\alpha\right\}$$

$$= \begin{cases} \dfrac{r(\beta)^2\gamma}{\gamma - \inf_\vartheta\|\beta - \vartheta\|_p^2} & \text{if } \inf_\vartheta\|\beta - \vartheta\|_p^2 \leq \gamma, \\ +\infty & \text{otherwise.} \end{cases}$$

Then the minimization objective can be simplified as

$$\inf_{\beta\in\mathbb{R}^d}\min_{\gamma\geq 0}\left\{\gamma\delta + \frac{1}{n}\sum_{i=1}^N\phi_\gamma(x_i,y_i;\beta)\right\}$$

$$= \inf_\beta\inf_{\gamma\geq\inf_\vartheta\|\beta-\vartheta\|_p^2}\left\{\gamma\delta + \frac{1}{n}\sum_{i=1}^N\frac{r_i(\beta)^2\gamma}{\gamma - \inf_\vartheta\|\beta - \vartheta\|_p^2}\right\}$$

$$= \inf_\beta\inf_{\gamma\geq\inf_\vartheta\|\beta-\vartheta\|_p^2}\left\{\gamma\delta + \text{MSE}(\beta)\frac{\gamma}{\gamma - \inf_\vartheta\|\beta - \vartheta\|_p^2}\right\}$$

$$= \inf_\beta\left(\sqrt{\text{MSE}(\beta)} + \sqrt{\delta}\inf_\vartheta\|\beta - \vartheta\|_p\right)^2,$$

where the last equality follows because $\gamma\delta + \frac{1}{n}\text{MSE}(\beta)\frac{\gamma}{\gamma - \inf_\vartheta\|\beta - \vartheta\|_p^2}$ is a convex function in $\gamma$ that tends to $+\infty$ approaching the boundaries $\inf_\vartheta\|\beta - \vartheta\|_p^2$ and $+\infty$, so the optimization over $\gamma$ can be solved using first-order condition. Then by Proposition 2.1, strong duality holds and,

$$\inf_\beta\sup_{\mathbb{P}:\mathcal{D}_{c_2}(\mathbb{P},\mathbb{P}_n)\leq\delta}\mathbb{E}_\mathbb{P}\left[(Y - \beta^\mathsf{T}X)^2\right] = \inf_{\beta,\vartheta}\left(\sqrt{\text{MSE}(\beta)} + \sqrt{\delta}\|\beta - \vartheta\|_p\right)^2.$$

This reduces the infinite-dimensional optimization to a finite-dimensional problem, where we interchanged $\inf_\vartheta$ and the quadratic function, since the quadratic function is monotone increasing on the positive reals. $\square$

The next proof is to Theorem 3.5 with the weak transferring cost function $c_{2,\lambda}\big((x,y),(u,v)\big) = \|x - u\|_2^2 + \lambda(\theta^\mathsf{T}x - \theta^\mathsf{T}u)^2 + \infty\cdot|y - v|$ with some $\lambda > 0$. The statements generalizes to multi-sites by first considering orthogonalizing the prior knowledge $\{\theta_1,\ldots,\theta_M\}$.

***Proof of Theorem 3.5.*** Following the proof of Theorem 3.3, we solve the optimization problem

$$\sup_{\Delta\in\mathbb{R}^d}\big((\beta^\mathsf{T}\Delta)^2 - 2r(\beta)\beta^\mathsf{T}\Delta - \gamma\|\Delta\|_2^2 - \gamma\lambda(\theta^\mathsf{T}\Delta)^2\big),$$

where we recall that $\gamma$ is the dual-variable in statement of Proposition 2.1, $\lambda > 0$ is the transferring strength, $\theta \in \mathbb{R}^d$ is the prior knowledge, and $r(\beta) = (y - \beta^\mathsf{T}x)^2$ is the residual in $\beta$.

Then let $\mathbb{O}$ be an orthogonal matrix, whose first column is $\theta/\|\theta\|_2$, then use $\widetilde{\Delta} := \mathbb{O}^{-1}\Delta$. The objective function now becomes

$$(\beta^\mathsf{T}\mathbb{O}\widetilde{\Delta})^2 - 2r(\beta)\beta^\mathsf{T}\mathbb{O}\widetilde{\Delta} - \gamma\|\widetilde{\Delta}\|_2^2 - \gamma\lambda\|\theta\|_2^2\widetilde{\Delta}_1^2,$$

where the last term follows because $\theta^\mathsf{T}\mathbb{O} = (\|\theta\|_2, 0, \ldots, 0)$, and $\widetilde{\Delta}_1$ denotes the first component of $\widetilde{\Delta}$. Now define

$$D = \mathrm{diag}\left\{\sqrt{\lambda\|\theta\|_2^2 + 1}, 1, \ldots, 1\right\},$$

and consider the change of variable $\bar{\Delta} = D\widetilde{\Delta}$, then the last two terms become

$$\|\widetilde{\Delta}\|_2^2 + \lambda\|\theta\|_2^2\widetilde{\Delta}_1^2 = \|D^{-1}\bar{\Delta}\|_2^2 + \lambda\|\theta\|_2\frac{\bar{\Delta}_1^2}{\lambda\|\theta\|_2^2 + 1} = \sum_{i=1}^{d}\bar{\Delta}_d^2 = \|\bar{\Delta}\|_2^2.$$

Therefore, the objective becomes

$$\begin{aligned}
&\sup_{\bar{\Delta}}\left((\beta^\mathsf{T}\mathbb{O}D^{-1}\bar{\Delta})^2 - 2r(\beta)\beta^\mathsf{T}\mathbb{O}D^{-1}\bar{\Delta} - \gamma\|\bar{\Delta}\|_2^2\right)\\
=&\sup_{\bar{\Delta}}\left(\|\beta^\mathsf{T}\mathbb{O}D^{-1}\|_2^2\|\bar{\Delta}\|_2^2 - 2r(\beta)\|\beta^\mathsf{T}\mathbb{O}D^{-1}\|_2\|\bar{\Delta}\|_2 - \gamma\|\bar{\Delta}\|_2^2\right)\\
=&\sup_{\bar{\Delta}}\left((\|\beta\|_{\Psi_\lambda} - \gamma)\|\bar{\Delta}\|_2^2 - 2r(\beta)\|\beta\|_{\Psi_\lambda}\|\bar{\Delta}\|_2\right)
\end{aligned}$$

which has finite optimal value $\dfrac{r(\beta)^2\|\beta\|_{\Psi_\lambda}^2}{\gamma - \|\beta\|_{\Psi_\lambda}^2}$ whenever $\gamma \geq \|\beta\|_{\Psi_\lambda}$, with $\Psi_\lambda$ denoting the positive-definite symmetric matrix,

$$\Psi_\lambda = I_d - \frac{1}{\|\theta\|_2^2 + \lambda^{-1}}\theta\theta^\mathsf{T},$$

that is independent of the choice of $\mathbb{O}$. The first equality follows because we applied Cauchy-Schwarz inequality and since $\bar{\Delta} \in \mathbb{R}^d$ is free, there is some $\bar{\Delta}$ that achieves equality. The rest of the proof follows exactly along the proof of Theorem 3.3 by completing the optimization over the dual problem using Proposition 2.1. $\qquad\square$

## C. Proof of Results in Classification.

**Lemma C.1.** *Consider the convex function $h_\beta(x) : \mathbb{R}^d \to \mathbb{R}$ by $x \in \mathbb{R}^d \mapsto \log\left(1 + \exp\left(-\beta^\mathsf{T}x\right)\right)$, for some $q > 0$ and $x' \in \mathbb{R}$. Then for every $\gamma > 0$, the constraint optimization problem $H_\beta(x')$ defined as,*

$$\begin{aligned}
&\sup_{x\in\mathbb{R}^p}\quad h_\beta(x) - \gamma\|x' - x\|_q,\\
&\text{s.t}\quad \theta^\mathsf{T}(x' - x) = 0,
\end{aligned}$$

*has optimal objective value,*

$$H_\beta(x') = \begin{cases} h_\beta(x') & \text{if } \inf_{\kappa\in\mathbb{R}}\|\beta - \kappa\theta\|_p \leq \gamma,\\ +\infty & \text{otherwise,} \end{cases}$$

*where $p, q \in [1, \infty)$ with $p^{-1} + q^{-1} = 1$.*

*Proof.* This lemma is a simple extension of (Shafieezadeh-Abadeh et al., 2015, Lemma 1). Following their proof, it is shown that

$$h_\beta(x) = h_\beta^{**}(x) = \sup_{0\leq\alpha\leq 1}\left((\alpha\beta)^\mathsf{T}x - \bar{h}^*(\alpha)\right),$$

where

$$\bar{h}^*(\alpha) = \begin{cases} \alpha\log(\alpha) + (1 - \alpha)\log(1 - \alpha) & \text{if } \alpha \in [0, 1],\\ +\infty & \text{otherwise,} \end{cases}$$

is the convex conjugate of the function $\log\left(1 + e^{-x}\right)$ (Proposition E.2). Then it is shown that the objective $H_\beta$ must has representation

$$\sup_{0 \le \alpha \le 1} \inf_{\|q\|_p \le \gamma} \sup_x \left((\alpha\beta + q)^\top x - \bar{h}^*(\alpha) - q^\top x'\right),$$

$$\text{s.t } \theta^\top(x - x') = 0.$$

Fixing $\alpha$ and $q$, then the inner maximization in $x$

$$\sup_x \left((\alpha\beta + q)^\top x - q^\top x'\right),$$

$$\text{s.t } \theta^\top(x - x') = 0,$$

has solution $(\alpha\beta)^\top x'$ subject to $\alpha\beta + q = \mu\theta$ for some $\mu \in \mathbb{R}$ derived using the first-order condition of the Lagrangian duality or $+\infty$ otherwise. Then condition on $\{\alpha\beta + q = \mu\theta | \mu \in \mathbb{R}\}$, the objective has representation

$$H_\beta(x') = \sup_{0 \le \alpha \le 1} \inf_{\|q\|_p \le \gamma} \left((\alpha\beta)^\top x' - \bar{h}^*(\alpha)\right) \text{ s.t } q = \mu\theta - \alpha\beta$$

$$= \sup_{0 \le \alpha \le 1} \inf_{\mu, \|\mu\theta - \alpha\beta\|_p \le \gamma} \left((\alpha\beta)^\top x' - \bar{h}^*(\alpha)\right).$$

Consider the constraint $\|\mu\theta - \alpha\beta\|_p \le \gamma$ over $\mu$. Suppose $\alpha > 0$, then dividing by $-\alpha$, we get the equivalent constraint $\left\{|\alpha| \left\|\beta - \frac{\mu}{\alpha}\theta\right\|_p\right\} \le \gamma$ over $\mu$. Defining the change of variable $\kappa := \frac{\mu}{\alpha}$, then since the Lagrange multiplier $\mu \in \mathbb{R}$ is free, we have $\kappa$ is free, and the constraint becomes $\inf_\kappa |\alpha| \|\beta - \kappa\theta\|_p \le \gamma$ over $\kappa \in \mathbb{R}$. If $\alpha = 0$, then $\inf_\mu \|\mu\theta - 0\| = 0 = 0 \cdot \inf_\kappa \|\beta - \kappa\theta\|_p$. So the equivalent constraint $\inf_\kappa |\alpha| \|\beta - \kappa\theta\|_p \le \gamma$ is valid for all $\alpha \in [0, 1]$. Then condition on $\{\alpha\beta + q = \mu\theta | \mu \in \mathbb{R}\}$, the objective becomes,

$$H_\beta(x') = \sup_{0 \le \alpha \le 1} \left((\alpha\beta)^\top x' - \bar{h}^*(\alpha)\right) \text{ s.t } \sup_{0 \le \alpha \le 1} |\alpha| \inf_\kappa \|\beta - \kappa\theta\|_p \le \gamma,$$

$$= \sup_{0 \le \alpha \le 1} \left((\alpha\beta)^\top x' - \bar{h}^*(\alpha)\right) \text{ s.t } \inf_\kappa \|\beta - \kappa\theta\|_p \le \gamma,$$

Recognizing that

$$\sup_{0 \le \alpha \le 1} \left((\alpha\beta)^\top x' - \bar{h}^*(\alpha)\right) = \sup_{0 \le \alpha \le 1} \alpha(\beta^\top x') - \bar{h}^*(\alpha) = \bar{h}^{**}(\beta^\top x') = h_\beta(x'),$$

we get

$$H_\beta(x') = \begin{cases} h_\beta(x') & \text{if } \inf_\kappa \|\beta - \kappa\theta\|_p \le \gamma, \\ +\infty & \text{otherwise.} \end{cases}$$

$\square$

The above Lemma C.1 is easily generalized to incorporate multiple orthogonality constraints $\{\theta_m^\top(x' - x) = 0 \,; m \in [M]\}$ using the exact same Lagrangian formulation. Again, recall $\Theta = \text{span}\{\theta_1, \ldots, \theta_M\}$. Thus the optimal objective value under multiple constraints becomes

$$H_\beta(x') = \begin{cases} h_\beta(x') & \text{if } \inf_{\vartheta \in \Theta} \|\beta - \vartheta\|_p \le \gamma, \\ +\infty & \text{otherwise.} \end{cases}$$

We now give the proof to Theorem 3.6.

***Proof of Theorem 3.6 for Logistic Loss.*** Using Proposition 2.1, we apply the strong duality, and consider the inner optimization problem

$$\sup_{\mathbb{P}:\mathcal{D}_{c_1,\infty}(\mathbb{P},\mathbb{P}_n) \le \delta} \mathbb{E}_\mathbb{P}\left[\log\left(1 + e^{-Y\beta^\top X}\right)\right] = \begin{cases} \inf_{\gamma \ge 0} \left\{\gamma\delta + \frac{1}{n}\sum_{i=1}^N \sup_{u \in \mathbb{R}^d} \left(\log\left(1 + e^{-y_i\beta^\top u}\right) - \gamma\|x_i - u\|_q\right)\right\}, \\ \text{s.t } \theta_m^\top(x_i - u) = 0, \text{ for all } m \in [M] \text{ and } i \in [N]. \end{cases}$$

For each $i \in [N]$, we apply Lemma C.1 to the maximization problem,

$$H_\beta(x_i) = \begin{cases} \sup_{u \in \mathbb{R}^d} \left( \log \left( 1 + e^{-y_i \beta^\intercal u} \right) - \gamma \| x_i - u \|_q \right), \\ \text{s.t } \theta_m^\intercal (x_i - u) = 0, \text{ for all } m \in [M]. \end{cases}$$

which has solution

$$\begin{cases} \log \left( 1 + e^{-y_i \beta^\intercal x_i} \right) & \text{if } \inf_{\vartheta \in \Theta} \| \beta - \vartheta \|_p \leq \gamma, \\ +\infty & \text{otherwise.} \end{cases}$$

Therefore, the maximization problem $\sup_{\mathbb{P}:\mathcal{D}_{c_1,\infty}(\mathbb{P},\mathbb{P}_n) \leq \delta} \mathbb{E}_\mathbb{P} \left[ \log \left( 1 + e^{-Y \beta^\intercal X} \right) \right]$ is bounded from above if and only if $\gamma \geq \inf_\vartheta \| \beta - \vartheta \|_p$. Under this condition, this reduces the inner optimization problem,

$$\sup_{\mathbb{P}:\mathcal{D}_{c_1,\infty}(\mathbb{P},\mathbb{P}_n) \leq \delta} \mathbb{E}_\mathbb{P} \left[ \log \left( 1 + e^{-Y \beta^\intercal X} \right) \right] = \inf_{\gamma \geq \inf_\vartheta \| \beta - \vartheta \|_p} \left\{ \gamma \delta + \frac{1}{n} \sum_{i=1}^N \log \left( 1 + e^{-y_i \beta^\intercal x_i} \right) \right\}$$

$$= \frac{1}{n} \sum_{i=1}^N \log \left( 1 + e^{-y_i \beta^\intercal x_i} \right) + \delta \inf_\vartheta \| \beta - \vartheta \|_p.$$

This concludes the proof. $\qquad\qquad\square$

We now give the proof to the maximum margin classifier using the hinge loss.

***Proof of Theorem 3.6 for Hinge Loss.*** As in the case to the proof of Theorem 3.3, we first consider the relaxed cost function

$$c_1((x,y),(u,v)) = \| x - u \|_q + \infty \cdot |y - v| + \lambda \cdot \sum_{m=1}^M |\theta_m^\intercal x - \theta_m^\intercal u|,$$

where we relaxed the transferring strength from $+\infty$ to some finite value $\lambda > 0$. We will then let $\lambda \to +\infty$. Again, by strong duality, we can solve the worst case hinge loss by solving the dual problem

$$\inf_{\gamma \geq 0} \left\{ \gamma \delta + \frac{1}{n} \sum_{i=1}^N \sup_u \left( (1 - y_i \beta^\intercal u)^+ - \gamma \| u - x_i \|_q - \gamma \lambda \sum_{m=1}^M |\theta_m^\intercal (x_i - u)| \right) \right\}.$$

Let $\Delta := u - x$, then we have

$$\sup_u \left( (1 - y \beta^\intercal u)^+ - \gamma \| u - x \|_q - \gamma \lambda \sum_{m=1}^M |\theta_m^\intercal (x - u)| \right)$$

$$= \sup_\Delta \left( (1 - y \beta^\intercal (\Delta + x))^+ - \gamma \| \Delta \|_q - \gamma \lambda \sum_{m=1}^M |\theta_m^\intercal \Delta| \right)$$

$$= \sup_\Delta \sup_{0 \leq \alpha \leq 1} \left( \alpha(1 - y \beta^\intercal (\Delta + x)) - \gamma \| \Delta \|_q - \gamma \lambda \sum_{m=1}^M |\theta_m^\intercal \Delta| \right)$$

$$= \sup_{0 \leq \alpha \leq 1} \sup_\Delta \left( -\alpha y \beta^\intercal \Delta - \gamma \| \Delta \|_q - \gamma \lambda \sum_{m=1}^M |\theta_m^\intercal \Delta| + \alpha(1 - y \beta^\intercal x) \right).$$

Where in the second equality we used $x^+ = \sup_{0 \leq \alpha \leq 1} \alpha x$. Fixing $\alpha$, consider the inner minimization in $\Delta$,

$$\sup_\Delta \left( -\alpha y \beta^\intercal \Delta - \gamma \| \Delta \|_q - \gamma \lambda \sum_{m=1}^M |\theta_m^\intercal \Delta| \right) = g^*(-\alpha y \beta),$$

where $g^*(\Delta^*)$ is the convex conjugate of $g(\Delta) := \gamma\|\Delta\|_q + \gamma\lambda\sum_{m=1}^M |\theta_m^\mathsf{T}\Delta|$. Set $\gamma\|\Delta_1\|_q =: g_1(\Delta_1)$ and $\gamma\lambda\sum_{m=1}^M |\theta_m^\mathsf{T}\Delta_2| =: g_2(\Delta_2)$, then by the *infimal convolution property* of convex conjugates (Theorem E.5), we know that

$$g^*(\Delta^*) = \inf_{\Delta_1^* + \Delta_2^* = \Delta^*} \left(g_1^*(\Delta_1^*) + g_2^*(\Delta_2^*)\right).$$

From Lemma B.2, we know that if $g^*(\Delta^*)$ is finite, then $g_2^*(\Delta_2^*) = 0$ subject to $\Delta_2^* = \sum_{m=1}^M \alpha_m\theta_m$ and $|\alpha_m| \le \lambda\gamma$ for all $m \in [M]$. Now it is well known that (Proposition E.2),

$$g_1^*(\Delta_1^*) = (\gamma\| \cdot \|_q)^*(\Delta_1^*) = I_{\{\|\Delta_1^*\|_p \le \gamma\}}(\Delta_1^*),$$

where $I_C(x)$ denotes the convex indicator on the set $C$. Therefore, letting $\lambda \to \infty$, the constraints $\{|\alpha_m| \le \lambda\gamma | m \in [M]\}$ is redundant, and we have

$$g^*(\Delta^*) = \begin{cases} 0 & \text{if } \inf_{\vartheta\in\Theta} \|\Delta^* - \vartheta\|_p \le \gamma, \\ +\infty & \text{otherwise,} \end{cases}$$

where we let $\Theta := \text{span}\{\theta_1, \ldots, \theta_M\}$. Therefore, $g^*(-\alpha y\beta)$ is finite if and only if $\inf_\vartheta \| - \alpha y\beta - \vartheta\|_p \le \gamma$. Now $y = \pm 1$, so we can remove $-y$, and this leaves us the condition that $\inf_\vartheta \|\alpha\beta - \vartheta\|_p \le \gamma$, which is equivalent to $\alpha\inf_\vartheta \|\beta - \vartheta\|_p \le \gamma$ for all $\alpha \in [0,1]$, including $\alpha = 0$. Taking supremum over $\alpha \in [0,1]$, the final condition is $\inf_\vartheta \|\beta - \vartheta\|_p \le \gamma$. Therefore, assuming the dual problem is bounded from above, it reduces as

$$\sup_{0\le\alpha\le 1} \sup_\Delta \left(-\alpha y\beta^\mathsf{T}\Delta - \gamma\|\Delta\|_q - \gamma\lambda\sum_{m=1}^M |\theta_m^\mathsf{T}\Delta| + \alpha(1 - y\beta^\mathsf{T}x)\right)$$

$$= \sup_{0\le\alpha\le 1} \left(I_{\{\inf_\vartheta \|\beta - \vartheta\|_p \le \gamma\}} + \alpha(1 - y\beta^\mathsf{T}x)\right)$$

$$= (1 - y\beta^\mathsf{T}x)^+ \quad \text{given} \quad \inf_\vartheta \|\beta - \vartheta\|_p \le \gamma.$$

Finally, the dual form of the distributionally robust optimization problem is

$$\inf_\beta \inf_{\gamma\ge 0} \left\{\gamma\delta + \frac{1}{n}\sum_{i=1}^N \sup_u \left((1 - y_i\beta^\mathsf{T}u)^+ - \gamma\|u - x_i\|_q - \gamma\lambda\sum_{m=1}^M |\theta_m^\mathsf{T}(x_i - u)|\right)\right\}$$

$$= \inf_\beta \inf_{\gamma\ge\inf_\vartheta \|\beta - \vartheta\|_p} \left\{\gamma\delta + \frac{1}{n}\sum_{i=1}^N (1 - y_i\beta^\mathsf{T}x_i)^+\right\}$$

$$= \inf_{\beta,\vartheta} \left\{\frac{1}{n}\sum_{i=1}^N (1 - y_i\beta^\mathsf{T}x_i)^+ + \delta\|\beta - \vartheta\|_p\right\}.$$

This completes the proof. $\qquad\square$

## D. Proof of Results in Mahalanobis Norm Regularization

***Proof of Corollary 3.7.*** This is a direct consequence of the convex conjugate of $\|x\|_\Lambda^2$ given in Proposition E.6. $\qquad\square$

Define the cost function $c_{1,\infty}^\Lambda\left((x,y),(u,v)\right) := \|x - u\|_\Lambda + \infty \cdot |y - v| + \infty \cdot \sum_{m=1}^M |\theta_m^\mathsf{T}x - \theta_m^\mathsf{T}u|$.

**Corollary D.1** (Theorem 3.6). *Suppose the loss function $\ell(X, Y; \beta)$ is either the logistic loss $\log\left(1 + e^{-Y\beta^\mathsf{T}X}\right)$ or the hinge loss $(1 - Y\beta^\mathsf{T}X)^+$, then for any $\Lambda \in \mathbb{S}_+^{d\times d}$ we have*

$$\inf_{\beta\in\mathbb{R}^d} \sup_{\mathbb{P}:\mathcal{B}_\delta(\mathbb{P}_N^X; c_{1,\infty}^\Lambda)} \mathbb{E}_\mathbb{P}\left[\ell(X, Y; \beta)\right]$$

$$= \inf_{\beta\in\mathbb{R}^d, \vartheta\in\Theta} \frac{1}{N}\sum_{i=1}^N \ell(x_i, y_i; \beta) + \delta\|\beta - \vartheta\|_{\Lambda^{-1}}.$$

*Proof.* For the logistic loss case, this is a direct consequence of the dual norm of $\|x\|_\Lambda$, for the hinge loss case this is a direct consequence of the convex conjugate of $\|x\|_\Lambda$. Both given by Proposition E.6. $\qquad\square$

# E. Useful Results on Convex Conjugation

In this section we review some results on the concept of convex conjugates that repeatedly come up in the proofs. For more details on convex conjugations, the interested readers can consult (Rockafellar, 1970, Section 12 & 16).

**Definition E.1** (Convex Conjugate). Let $f : \mathbb{R}^d \to \mathbb{R}$ be a real-valued function on the Euclidean space, then the convex conjugate of $f$ is the function $f^* : \mathbb{R}^d \to \mathbb{R}$ that evaluates $x^* \in \mathbb{R}^n$ by

$$f^*(x^*) = \sup_{x \in \mathbf{dom}(f)} \left( x^{*\top}x - f(x) \right).$$

This is also called the *Legendre transformation* of $f$, and the *Legendre-Fenchel transformation* for $f$ defined on arbitrary real topological vector spaces. Here we collect some examples of convex conjugates that appeared in the appendix. These are well-known.

**Proposition E.2.** *Let $p, q \geq 1$ such that $\dfrac{1}{p} + \dfrac{1}{q} = 1$.*

1. *The convex conjugate of the absolute value function $f(x) = |x|$ on $\mathbb{R}$ is given by $|\cdot|^*(x^*) = I_{|x^*| \leq 1}(x^*)$, the convex indicator function on the set $\{|x^*| \leq 1 | x^* \in \mathbb{R}\}$.*

2. *The convex conjugate of the q-norm $\|x\|_q$ on $\mathbb{R}^d$ is given by $\|\cdot\|_q^*(x^*) = I_{\|x^*\|_p \leq 1}(x^*)$, the convex indicator function on the set $\{\|x^*\|_p \leq 1 | x^* \in \mathbb{R}^d\}$.*

3. *The convex conjugate of $\dfrac{1}{2}\|x\|_q^2$ on $\mathbb{R}^d$ is given by $\left( \dfrac{1}{2}\|\cdot\|_q^2 \right)^*(x^*) = \dfrac{1}{2}\|x^*\|_p^2$.*

4. *The convex conjugate of $\log\left(1 + e^{-x}\right)$ on $\mathbb{R}$ is given by*

$$\begin{cases} x^* \log(x^*) + (1 - x^*) \log(1 - x^*) & \text{if } x^* \in (0, 1) \\ 0 & \text{if } x^* = 0, 1 \\ +\infty & \text{otherwise}. \end{cases}$$

Another easy consequence from the definition of convex conjugation is the below scaling laws.

**Proposition E.3** (Scaling Laws). *Let $f^*(x^*)$ be the convex conjugate of $f(x)$ on $\mathbb{R}^d$. Then we have,*

1. *the convex conjugate of $f(ax)$ whenever $a \neq 0$ is given by $f^*(x^*/a)$.*

2. *the convex conjugate of $af(x)$ whenever $a > 0$ is given by $af^*(x^*/a)$.*

Let $\Gamma\left(\mathbb{R}^d\right)$ denote the class of proper convex lower-semi continuous functions on $\mathbb{R}^d$, the next statement says that this conjugation induces an one-to-one symmetric correspondence on the class $\Gamma\left(\mathbb{R}^d\right)$. It is a cornerstone of modern convex analysis and used in the proof of Theorem 3.3 and Lemma C.1.

**Theorem E.4** (Fenchel-Moreau). *Let $f$ be a proper convex, lower semi-continuous function on $\mathbb{R}^d$, then*

1. *the convex conjugation $f \mapsto f^*$ is a bijection on $\Gamma\left(\mathbb{R}^d\right)$;*

2. *$f \in \Gamma\left(\mathbb{R}^d\right) \iff f^{**} = f$.*

*Proof.* For a proof please consult (Rockafellar, 1970, Section 12). $\square$

The next statement concerns the commutativity of convex conjugation and function summation. Its usefulness is profound, and applied to the proof of Theorem 3.3 and Theorem 3.6.

**Theorem E.5** (Infimal Convolution Property of Convex Conjugation). *Let $f_1, \ldots, f_M$ be proper convex functions on $\mathbb{R}^d$, then*

$$(f_1 \square \ldots \square f_M)^* = f_1^* + \ldots f_M^*,$$

*and*

$$(f_1 + \ldots + f_M)^*(x^*) = \inf_{x^*} \{f_1^*(x_1^*) + \ldots f_M^*(x_M^*) \mid x_1^* + \ldots + x_M^* = x^*\}.$$

*Proof.* For a proof please consult (Rockafellar, 1970, Theorem 16.4). □

**Proposition E.6.** *Let $\Lambda \in \mathbb{S}_+^{d \times d}$, then the dual norm of $\|x\|_\Lambda$ is $\|x\|_{\Lambda^{-1}}$. The Cauchy-Schwarz inequality $x^\top u \leq \|x\|_\Lambda \|u\|_{\Lambda^{-1}}$ holds, and equality is attainable. The convex conjugate of $\|x\|_\Lambda$ is given by $I_{\|x^*\|_{\Lambda^{-1}} \leq 1}(x^*)$, and the convex conjugate of $\|x\|_\Lambda^2$ is given by $\|x^*\|_{\Lambda^{-1}}^2/4$.*

*Proof.* The dual norm of $\|x\|_\Lambda$, the Cauchy-Schwarz inequality and attainability of equality follows from (Blanchet et al., 2019b, Lemma 1). Now to compute the convex conjugate of $\|x\|_\Lambda^2$, we want to evaluate

$$\sup_{x \in \mathbb{R}^d} (x^{*\top} x - \|x\|_\Lambda^2).$$

By the Cauchy-Schwarz inequality we have $x^{*\top} x \leq \|x\|_\Lambda \|x^*\|_{\Lambda^{-1}}$, and so we have

$$x^{*\top} x - \|x\|_\Lambda^2 \leq \|x\|_\Lambda \|x^*\|_{\Lambda^{-1}} - \|x\|_\Lambda^2.$$

Hence

$$\sup_{x \in \mathbb{R}^d} (x^{*\top} x - \|x\|_\Lambda^2) \leq \sup_{t \geq 0}(t\|x^*\|_{\Lambda^{-1}} - t^2) = \frac{1}{4}\|x^*\|_{\Lambda^{-1}}^2.$$

By attainability of equality in the Cauchy-Schwarz inequality, the supremum are equal, and we have

$$\sup_{x \in \mathbb{R}^d} (x^{*\top} x - \|x\|_\Lambda^2) = \frac{1}{4}\|x^*\|_{\Lambda^{-1}}^2.$$

This proofs the convex conjugate of $\|x\|_\Lambda^2$. Now consider the convex conjugate of $\|x\|_\Lambda$, then we need to evaluate

$$\sup_{x \in \mathbb{R}^d} (x^{*\top} x - \|x\|_\Lambda),$$

again, by Cauchy-Schwarz and the attainability of equality, we have

$$\sup_{x \in \mathbb{R}^d} (x^{*\top} x - \|x\|_\Lambda) = \sup_{x \in \mathbb{R}^d} (\|x\|_\Lambda \|x^*\|_{\Lambda^{-1}} - \|x\|_\Lambda)$$

$$= \sup_{x \in \mathbb{R}}(\|x\|_\Lambda (\|x^*\|_{\Lambda^{-1}} - 1))$$

$$= \begin{cases} 0 & \text{if } \|x^*\|_{\Lambda^{-1}} \leq 1, \\ +\infty & \text{otherwise.} \end{cases}$$

This completes the proof.

□

# F. Toland's Duality

The duality theory of Toland's (Toland, 1978; 1979) concerns the minimization of nonconvex functions, in particular, applies to the minimization of the difference of convex functions (DC problems). The duality holds under minimal conditions, and one tries to see if the DC problem can be transformed into something more manageable.

**Theorem F.1** (Toland's Duality). *Let $f$ and $g$ be functions on $\mathbb{R}^d$, if $f \in \Gamma(\mathbb{R}^d)$, then we have*

$$\inf_{x \in \mathbb{R}^d} \{f(x) - g(x)\} = \inf_{x^* \in \mathbb{R}^d} \{g^*(x^*) - f^*(x^*)\}.$$

Toland's duality is implicitly used in the proof to Theorem 3.3 and Lemma C.1 which also sketches a proof to the above duality theorem.

