# OpenReview forum: "Knowledge-Guided Wasserstein Distributionally Robust Optimization"
_ICML.cc/2025/Conference — ICML 2025 poster_

### Official Review · Reviewer_85zy · 2025-03-10

**Overall Recommendation:** 3

**Summary:**

This paper investigates distributionally robust optimization (DRO), focusing on Wasserstein distance-based DRO (W-DRO) while introducing a novel knowledge-guided cost function to further enhance the robustness and performance of DRO frameworks. The authors provide an extensive and thorough review of DRO, elaborating on its properties, and specifically analyzing the mathematical underpinnings of W-DRO. Building on this theoretical foundation, the authors propose a Knowledge-Guided Transport Cost (KGTC), which incorporates knowledge coefficients that reflect prior knowledge about the data or task to guide the transport cost within the Wasserstein ball. By embedding these knowledge-guided adjustments, the model aims to achieve better generalization and robustness to distributional shifts. The proposed method is validated theoretically and empirically on linear regression and binary classification tasks, demonstrating promising improvements over existing methods.

**Claims And Evidence:**

This is a rigorous paper, most of the claims are backed with theoretical support. Only more intuition would be better.

**Essential References Not Discussed:**

No additional references needed.

**Experimental Designs Or Analyses:**

I have check the Experimental designs, which are quite reasonable.

**Methods And Evaluation Criteria:**

No.

**Other Comments Or Suggestions:**

Please see the weaknesses.

**Other Strengths And Weaknesses:**

Strengths
- Theoretical Rigor and Solid Analysis:
  - The paper presents strong theoretical foundations with rigorous and logical arguments supporting the development of the Knowledge-Guided Transport Cost (KGTC).
  - The analytical treatment of both standard DRO and Wasserstein DRO is comprehensive, offering valuable insights into the properties and limitations of existing approaches.
  - The extension toward knowledge-guided variants is well-motivated from a theoretical perspective, making a notable contribution to the DRO literature.
- Comprehensive Coverage of Use Cases:
  - The authors consider multiple scenarios, including both strong and weak transfer settings, showcasing the generality and adaptability of their proposed framework.
  - The method is applied to both regression (linear regression) and classification (binary classification)tasks, which highlights its versatility and potential for broader applications.
- Promising Empirical Results:
  - The numerical experiments support the theoretical claims, demonstrating significant performance gains over standard DRO formulations.
  - The improvements observed in diverse tasks highlight the practical potential of incorporating knowledge-guided components into DRO.

Weaknesses and Areas for Improvement
- Lack of Discussion on Convergence and Statistical Guarantees:
  - One notable omission is the absence of a discussion regarding convergence rates and statistical guaranteesof the proposed method.
  - While the method is shown to be effective in terms of robustness and performance, readers would benefit from understanding whether there are provable bounds on convergence speed or finite-sample guaranteesthat back up the empirical observations.
- Intuition Behind Knowledge-Guided Cost Function Needs Clarification:
  - Although the knowledge-guided cost function is mathematically well-defined, there is a lack of intuitive explanation regarding why and how incorporating multiple knowledge coefficients leads to better transport cost estimation and performance gains.coefficients would help clarify the practical utility of the approach.
- Practical Applicability and Generalization to Real-World Scenarios:
  - While the theoretical and empirical studies are strong, the real-world applicability of the knowledge-guided cost function is underexplored.
  - It would be beneficial to provide concrete real-world scenarios or case studies where domain knowledge is readily available and can be leveraged for improved robustness (e.g., healthcare, finance, or autonomous systems).
- Computational Complexity and Overhead Not Addressed:
  - A major concern is the potential computational overhead introduced by incorporating the knowledge-guided cost function.
  - Intuitively, the use of knowledge coefficients and the corresponding adjustments in transport cost computation could lead to higher space and time complexity, especially in large-scale problems or high-dimensional settings.
  - The paper does not discuss the computational implications, nor does it provide empirical measurements(e.g., runtime, memory usage) to reassure readers that the method remains computationally feasible.
  - Adding an analysis or even a simple comparison of computational efficiency between standard W-DRO and KGTC-enhanced W-DRO would significantly strengthen the practical relevance of the paper.

**Questions For Authors:**

N/A

**Relation To Broader Scientific Literature:**

This paper enhances the W-DRO with a novel regularization, which enhances the performance under extreme situations.

**Theoretical Claims:**

I didn't the details of the proofs.

---

> ### Author Rebuttal · Authors · 2025-03-31
>
> We sincerely thank the reviewer for the encouraging and detailed feedback. Here we list our responses to the weaknesses and questions suggested by the reviewer.
>
> **W1: Lack of Convergence and Guarantees**
>
> We acknowledge the absence of an explicit discussion on convergence rates and statistical guarantees in this paper. However, the primary focus of our work is to establish the theoretical equivalence between shrinkage-based transfer learning and the Wasserstein Distributionally Robust Optimization (WDRO) framework and the statistical properties are beyond the scope. This perspective provides a unified approach to analyzing transfer learning problems.
>
> Many prior studies on specific cases of our proposed method have already explored the convergence of the optimal estimator as the radius shrinks; e.g. [1]. We plan to leverage the WDRO framework to establish formal statistical guarantees in future work.
>
> **W2 I: Intuition on Cost Function**
>
> The standard WDRO cost function takes the form $c(x,x') = \Vert x-x'\Vert_q^2$, allowing perturbations of the covariate in all directions. However, if we believe that prior knowledge $\theta$ serves as a trustworthy proxy for $β^*$ (the true optimum of Problem (SO), Line 125, Right), then it is natural to minimize perturbations in the predictive direction of $\theta$. Specifically, we constrain the perturbed point $x'$ so that the discrepancy between the predictions of $x$ and $x'$ under $θ$ is small , i.e., $θ^\top x' ≈θ^\top x$. Since $x$ is a point in the empirical measure, this ensures that the transported measure $P_N$ is mapped to distributions that preserve predictions under $θ$. Consequently, taking the worst-case over the ambiguity set yields estimators that perform at least as good as using $θ$ naively.
>
> **W2 II: Multi-Source Knowledge**
>
> In applications such as electronic health records, multiple clinical trials are conducted in different hospitals. These source domains typically represent majority but distinct populations. When the target domain involves a mixed-ethnicity population, it is reasonable to expect that the target estimator should be some combination of estimators derived from the source domains. This is naturally captured by our penalty term, for the case of two source data, this takes the form  $\inf_{κ_1,κ_2}\Vertβ-κ_1θ_1-κ_2θ_2\Vert_p,$ allowing the model to automatically search for the best combination of source knowledge. We refer to this as a “multi-source ensemble” of prior knowledge, where the learning process profiles and distills useful information from multiple sources.
>
> **W3: Practical Applications**
>
> Thanks for your advice.  To illustrate the applicability of our KG-WDRO framework in the real world, we apply it to the TransGLM dataset [2] on 2020 U.S election results at the county level.
>
> Counties are labeled as '1' if the Democrat candidate won, and 0 otherwise. We assess KG-WDRO vs. TransGLM by classifying county-level outcomes in eight target states, using data from the remaining states as source knowledge. The cleaned dataset includes 3111 counties and 761 standardized predictors across 49 states.
>
> Using 2100 counties as source, we predict results in eight target states (~100 counties each). KG-WDRO outperforms TransGLM in 5 of 8 states, reducing overall logloss by 7.6%, see table (https://figshare.com/s/e00c2d14f2c15ac02ed9). Both transfer learning methods significantly outperform the standard WDRO estimator. We will add this experiment to our main text.
>
> **W4: Computational Complexity**
>
> Theorem 3.2 transforms the infinite-dimensional problem (Line 240) into a tractable convex program (Line 242). Given the number $M$ of external sources is finite, the convex program takes the form $\inf_{β,κ}\Vert \mathbf{y}-\mathbf{X}β\Vert_2+\sqrt{δ}\Vertβ-κ_1θ_1-...-κ_Mθ_M\Vert_p,$ where $κ\in\mathbb{R}^M=[κ_1,...,κ_M]^\top$. Setting $κ=[0,...,0]^\top$, this reduces penalty term to a $p$-norm regularization, like Lasso or Ridge. When $κ$ is free, there are $M$ parameters. Consequently, the total number of parameters increases from $d$ (dimension of $\beta$) to $d+M$. As long as the number of knowledge sources remains finite—which is a reasonable assumption given the cost of experiments—the parameter size of the program remains of order $O(d)$ in high-dimensionality.
>
> Its computational complexity remains roughly the same as traditional regularization techniques. In our experiments, the KG-WDRO optimization problem is efficiently solved using CVXPY with Mosek. Additionally, we conduct an empirical study to demonstrate that their runtimes remain similar (https://figshare.com/s/9af171f8a0dcc3eb32da).
>
> [1] Blanchet, J., Murthy, K., & Si, N. (2022). Confidence regions in Wasserstein distributionally robust estimation. Biometrika, 109(2), 295-315.
>
> [2] Tian, Y., & Feng, Y. (2023). Transfer learning under high-dimensional generalized linear models. Journal of the American Statistical Association, 118(544), 2684-2697.

---

> > ### Comment · Reviewer_85zy · 2025-04-04
> >
> > Thanks for the detailed responses provided by the authors. Most of my concerns are addressed. Hence, I stay positive on the acceptance of this paper.

---

### Official Review · Reviewer_LRuU · 2025-03-11

**Overall Recommendation:** 3

**Summary:**

This work introduces a framework for transfer learning called Knowledge-Guided Wasserstein Distributionally Robust Optimization. In face of the overly conservative property of WDRO, the proposed framework adapts the Wasserstein ambiguity set using external knowledge (augment the transport cost function with an penalty term involving with prediction discrepancy).  They establish the equivalence between KG-WDRO and shrinkage-based estimation methods, and demonstrate the effectiveness of KG-WDRO in improving small-sample transfer learning through numerical simulations.

## update after rebuttal

After reading other reviews and the rebuttals, I decide to maintain my score.

**Claims And Evidence:**

Yes

**Essential References Not Discussed:**

I think the references are relatively sufficient.

**Experimental Designs Or Analyses:**

I checked the experimental part.

**Methods And Evaluation Criteria:**

Yes

**Other Comments Or Suggestions:**

No

**Other Strengths And Weaknesses:**

**Strengths:**

- Integrate prior knowledge into WDRO for linear regression and binary classification. Provide equivalence between KG-WDRO and shrinkage-based estimation method. Interprets a broad range of knowledge transfer learning approaches through the lens of distributional robustness.
- The perspective of the research is supported by theoretical advancements and empirical evidence.
- The experimental section presents extensive results, qualitative and quantitative.

**Weaknesses:**

* Lack of consistency in writing. It is hard to follow the connection between transfer learning and WDRO, the authors mention WDRO in the title, then suddenly mention transfer learning in the beginning of the abstract, without further discuss how the proposed framework benefits transfer learning.
* Lack of introduction of the notation before using them.
  * In Example 1.  What is the meaning of $\delta, \kappa$ ?
  * In section 2.1. What does the notation $\pi (A \times \mathbb{R}^d )$​ mean?
* Lack of justification and ablation experiment for the selection of hyperparameters $\delta$ and $\lambda$ .

**Questions For Authors:**

- In line 20, the authors mention "Our method constructs smaller Wasserstein ambiguity sets", the reader may want to acquire a quantitative analysis about it, can it be further discussed?
- The authors mention ''This framework mitigates the conservativeness of standard WDRO''. According to my understanding, the conservativeness of vanilla WDRO is due to worst-case optimization. And it seems to be relatively easier to bypass the worst case if one wants to alleviate the conservativeness of WDRO, do the authors demonstrate the challenge of directly using some vanilla methods (for example, linear interpolation between worst case and random case)?

**Relation To Broader Scientific Literature:**

N/A

**Theoretical Claims:**

I checked part of them.

---

> ### Author Rebuttal · Authors · 2025-03-31
>
> We sincerely thank the reviewer for the encouraging and detailed feedback.
>
> **W1: Inconsistent Writings**
>
> We acknowledge that the connection between WDRO and transfer learning may not be immediately clear. We will refine our writing to ensure a smoother and more intuitive transition between these two concepts.
>
> The main contribution of our paper is to establish that a broad class of shrinkage-based transfer learning objectives can be equivalently formulated as a WDRO problem. This provides a distributionally robust perspective on transfer learning. The title includes the phrase “knowledge-guided”, which refers to a type of transfer learning known as *domain adaptation*-adapting models trained on a source domain to perform well on a related target domain. In our framework, the prior knowledge is represented by the model parameters learned from the source domain, and the optimization in the target domain is guided by this prior knowledge. Applying the DRO principle allows us to rigorously derive a penalized estimation framework, while also ensuring robustness to distributional shifts.
>
> **W2: Notations**
>
> We thank the reviewer for pointing out these oversights. We will add the relevant definitions to the main text. The notation $δ\geq 0$ denotes the radius of the Wasserstein ball centered at the empirical measure. The free variable $κ\in\mathbb{R}$ in the optimization is interpreted as the projection coefficient of $β$ onto $θ$. In $π(A\times\mathbb{R}^d)$, $π$ denotes a probability on the product space $\mathbb{R}^d \times \mathbb{R}^d$, and $A$ is a Borel measurable subset of $\mathbb{R}^d$.
>
> **W3: Hyperparameter Tuning**
>
> We use cross validation (CV) to tune the hyperparameters $δ$ and $λ$. In the future, our goal is to develop an automated hyperparameter selection method, akin to the methods introduced in (Blanchet et al, 2019a)(Line 462, Left) and (Blanchet et al, 2022)(Line 476, Left), building on the theoretical framework of WDRO.
>
> For ablation study, we conducted experiments where we set $λ=0$ (no transfer), reducing KG-WDRO to a standard WDRO formulation. As shown in the upper two subfigures of Figure 1, our method consistently improves regression error and classification accuracy, when the correlation between the prior $θ$ and the true $β$ is as low as 0.3.
>
> In this revision, we include a new ablation study in high-dimensional regression setting, which will be added to main text. We fix the values of $δ$ to either $δ^*$, where $δ^*$ is tuned via CV on a standard WDRO estimator, or fixed to $δ=3$. Using fixed values, we fit KG-WDRO estimators across a grid of $λ$. The resulting out-of-sample (OOS) performances are plotted in the figure (https://figshare.com/s/05b069e330136338fa4e). When the correlation between the prior $θ$ and the true $β^*$ is high, setting $λ\to∞$ yields the best OOS performance. As the correlation decreases, smaller values of $λ$ lead to better results. This finding highlights the need to include $λ$ to control the extent of bias, and aligns with the intuition that stronger correlations warrant larger values of $λ$. Red dots in the plot represent the OOS performance obtained via CV. The $λ$-coordinates of these red dots follow the trend of the curves. The red dots lie above the curves, indicating improved performance when $δ$ is tuned.
>
> **Q1: Smaller Ambiguity Set**
>
> Yes. For the case when $λ=∞$, we can upper bound the minimax objective (Line 239, Left) by $\inf_{α\in\mathbb{R}} \mathbb{E}_{\mathbb{P}_N}(Y-(\alpha θ)^\top X)^2$, which is the in-sample risk of the prior $θ$ adapted on target data. In standard WDRO, this upper bound is trivially given by $\mathbb{E}(Y^2)$. We can show that the ambiguity set becomes strictly smaller when $λ= ∞$ compared to when $λ=0$. We will provide these detailed quantitative discussions on the ambiguity set in the new draft.
>
> **Q2: Mitigating Conservativeness**
>
> The goal of KG-WDRO is to ensure robustness against statistical noise while avoiding over conservativeness. One could simply use the empirical risk minimization solution, $β_{ERM}$, from Problem (ERM) (Line 133) to prevent over-conservatism but in a small-sample regime, computing $β_{ERM}$ is often infeasible without introducing bias, such as that induced by standard WDRO. Even in cases where $β_{ERM}$ is computable, its out-of-sample performance is typically poor due to the high uncertainty associated with limited data.
>
> Linear interpolation between the worst case and the random case may also fail. It would inherit both high uncertainty (from the random distribution) and high conservatism (from worst-case distribution), leading to a suboptimal trade-off between bias and variance. Overall, there is no free lunch if no additional information is provided. In contrast, our framework leverages prior knowledge in a structured manner to mitigate conservativeness while maintaining robustness, thereby offering a principled alternative to simple interpolation-based approaches.

---

### Official Review · Reviewer_UVsb · 2025-03-15

**Overall Recommendation:** 2

**Summary:**

The authors believe that traditional Wasserstein Distributionally Robust Optimization (WDRO) has a conservative tendency, which can lead to suboptimal performance. They argue that in real-world scenarios, prior knowledge can be leveraged to enhance model performance and robustness. Therefore, they propose that integrating prior knowledge into the Wasserstein Distributionally Robust Optimization framework remains an open question. In this work, the authors introduce a novel framework called Knowledge-Guided Wasserstein Distributionally Robust Optimization (KG-WDRO), which utilizes external knowledge (parameters) to adjust the Wasserstein ambiguity set by constraining the transportation costs along directions indicated by the prior knowledge. The authors summarize that this strategy allows the model to concentrate uncertainty in areas where the prior knowledge is less reliable, effectively enhancing the robustness of knowledge-guided generalization.

**Claims And Evidence:**

1. Introducing prior knowledge is a very good question, but the authors' explanation of it in the early part of the paper is quite vague.

**Essential References Not Discussed:**

n/a

**Experimental Designs Or Analyses:**

The numerical results presented need to incorporate more real-world scenarios for better applicability and relevance.

**Methods And Evaluation Criteria:**

Is this method still viable in high-dimensional knowledge scenarios, such as neural networks? Lack of more practical scenarios.

**Other Comments Or Suggestions:**

n/a

**Other Strengths And Weaknesses:**

Weakness:

1.Introducing prior knowledge is a very good question, but the authors' explanation of it in the early part of the paper is quite vague.

2. The authors use linear regression and binary classification problems to validate the proposed method. Is this appropriate? Linear regression and binary classification are not very complex problems, and there are already many methods to solve them. To validate KG-WDRO, the authors should provide problems that match its complexity.

3. There is an error in the formula on line 182; "inf" and "min" represent different concepts.

4. From Theorem 3.2, it appears to resemble a regularization method, making it difficult to determine from a theoretical perspective whether the improvement in performance is due to the prior knowledge or the method proposed by the authors. What is the impact of the amount of prior knowledge on performance improvement?

5. Is this method still viable in high-dimensional knowledge scenarios, such as neural networks?

**Questions For Authors:**

n/a

**Relation To Broader Scientific Literature:**

n/a

**Theoretical Claims:**

The theoretical framework is built on strong assumptions and has notable limitations. Additionally, the professors embrace significant hypotheses in their analyses, such as the definition of delta in Line 702. Why is delta defined in this particular manner?

---

> ### Author Rebuttal · Authors · 2025-03-31
>
> We sincerely thank the reviewer for the encouraging and detailed feedback.
>
> **W1: Explanation of Using Prior Knowledge.**
>
> Our transfer learning approach falls under *Domain Adaptation*, which adapts models trained on a source domain to perform well on a related target domain with limited labeled data.
>
> A key application is in clinical trials, where the binary outcome $Y\in\\{0,1\\}$ indicates treatment success or failure, and the high-dimensional covariate $X$ encodes a patient’s physical and health conditions along with treatment details. Data scarcity is common—especially for underrepresented populations. To address this, we leverage a classifier trained on a majority group (parameterized by $θ$) as a reference to estimate a classifier for the minority group (parameterized by $β$). This knowledge-guided transfer learning reduces uncertainty by anchoring the search for $β$ in the direction of $θ$. We will include this example in the new draft to clarify our setting.
>
> **W2: Model Complexity**
>
> We would like to emphasize that in our settings , due to the inherently small sample size or lack of labeled data in the target domain, it is necessary to limit the complexity of our machine learning models. For example, in clinical trials, there are usually only ~100 observations. Therefore, neural-network type methods may overfit.
>
> The tractable reformulation we obtained for generalized linear models and SVMs provides valuable insights into how shrinkage-based or penalized estimation for transfer learning objectives should look like. This distinguishes our approach from many previous works, as discussed in Table 1. Furthermore, as we will elaborate below, the techniques underlying KG-WDRO can be directly extended to more complex machine learning models, including neural networks.
>
> **W3: Typo**
>
> Thanks for catching this. We will correct the $\inf$ to $\min$ in the statement on (Line 182, Left).
>
> **W4: Connection to Regularization.**
>
> Yes, we acknowledge that the proposed method can be interpreted as a form of regularization, and in fact, this tractable reformulation is a key contribution of our paper. It allows us to efficiently solve the infinite-dimensional minimax problem as a convex program.
>
> This suggests that the performance improvement is directly attributable to the integration of prior knowledge into the cost function, which translates into a regularization/penalty term that encourages collinearity with the prior. Therefore, both the prior knowledge and our method play important roles. Regarding the impact of the quality of prior knowledge, the upper two subfigures in Figure 1 demonstrate that our method improves regression mean squared error  and classification accuracy even when the correlation between the prior knowledge and the true parameter is as low as 0.3, with hyperparameters selected via cross-validation.
>
> **W5: High-dimensional Models.**
>
> Our method remains applicable in high-dimensional settings. We can apply this principle to neural networks as follows: suppose we pre-trained a multilayer perceptron (MLP) on the source. We can fix all the hidden layers and treat the output layer’s parameters as prior knowledge, denoted by $θ$. Then, on the source domain, we fine-tune the MLP’s output layer using the KG-WDRO objective, effectively re-learning the output layer on the target domain but using the previous output layer as knowledge guidance. Mathematically, suppose the pretrained MLP is represented by $f(X) = θ^\top h(X)+b$, where $h:\mathbb{R}^d\to\mathbb{R}^k$ is the nonlinear hidden layers, then the KG-WDRO framework for the MLP on the target data is to solve the optimization problem $ \inf_{β,c,κ}\Vert \mathbf{y}-β^\top h(\mathbf{X})-c\Vert_2+\sqrt{\delta} \Vert[θ,b]-κ[β,c]\Vert_p,$ by taking the hidden layers $h(\cdot)$ as fixed.
>
> Following this, we conduct an additional experiment (https://figshare.com/s/29e12260c87f084eeb54) between KG-MLP versus naive MLP in a setup similar to Figure 2 of Main Text, which will be added to the new version of the paper. Specifically, we consider an 11-dimensional setting. The response $y$ is the sum of 5 nonlinear ($\sin, e^{-x^2}, \log(|x|+1), \tanh$ and an interaction) and 5 linear basis. The linear coefficients for source and target are generated using the same method in Figure 2 for different correlations. We use a 2-layer MLP with 11 and 10 nodes in each layer. The pretrained MLP was trained on 5,000 data points, then KG-MLP was fine-tuned on 130 target points, while the naive MLP used the same data. We see that KG-MLP outperforms naive MLP consistently, especially when correlation is high.
>
> **Lack of Real Data**
>
> We refer the reviewer to **W3** in the rebuttal to Reviewer 85zy to see the addition of the real data analysis.
>
> **$\Delta$ on Line 802**
>
> Here $\Delta$ is just a free variable of the $d$-dimensional space $\mathbb{R}^d$ to represent the perturbation $x-x'$ in the ambiguity set, we will refurbish the appendix for better exposition.

---

> > ### Comment · Reviewer_UVsb · 2025-04-09
> >
> > Thank you for the rebuttal. In the context of distribution optimization, introducing a prior parameter to facilitate a robust transition is not a novel approach. The initial configuration for parameter construction can be challenging to control, which is likely why the authors have introduced a new parameter, beta, to relieve this issue.
> >
> > The construction of 𝜃 and its varying control should be key to the idea. In optimization, this involves the variance question. It seems that the authors lack such analysis.
> >
> > Overall, I find that the optimization solution does not offer significant new insights. The use of parametric optimization with robust control is a well-established method. Furthermore, the results are primarily applied to a limited set of simple cases, lacking broader applications. As a result, I see few advancements for our ML community.
> >
> > ---------I apologize for not realizing that rebuttal discussions cannot be submitted through the official comments button; the author remains unseen.--------------------

---

### Official Review · Reviewer_RPYq · 2025-03-15

**Overall Recommendation:** 4

**Summary:**

The paper introduces a transfer-learning variant of Wasserstein Distributionally Robust Optimization. Given some external knowledge, which the authors represent by a vector $\theta$, they construct an ambiguity region based on a Wasserstein distance with $\theta$-dependent cost function. This makes the ambiguity region less pessimistic. They show that the resulting optimization problems are equivalent to several known forms of regularization.

## Update after rebuttal
I appreciate the authors response. My review remains unchanged.

**Claims And Evidence:**

The claims are supported by proofs and the approach is well-motivated.

**Essential References Not Discussed:**

-

**Experimental Designs Or Analyses:**

The experimental setup seems reasonable.

**Methods And Evaluation Criteria:**

The proposed methods are adequate.

**Other Comments Or Suggestions:**

-

**Other Strengths And Weaknesses:**

Strengths:
* The paper is very well written and pleasant to read
* The results are very elegant. The proposed criteria seem intractable, but are rewritten to more tractable-looking regularized problems.

Weaknesses:
* The proposed setup introduces an additional parameter $\lambda$ without any guidance on how to choose this value.

**Questions For Authors:**

Suppose the prior knowledge $\theta$ is *unhelpful* for the learning problem at hand, how much does this hamper the learning? Will the method, given sufficient data, still converge to the optimum? Does this depend on whether $\lambda=\infty$ or not?

When setting $\lambda=\infty$, we are basically just adding some constraints to the optimization problem. Can this really be considered transfer learning?

**Relation To Broader Scientific Literature:**

WDRO is a popular optimization framework that has been related to several forms of regularization. This work introduces a WDRO-variant for transfer learning and relates it to several regularization measures.

**Theoretical Claims:**

I did not check the proofs.

---

> ### Author Rebuttal · Authors · 2025-03-31
>
> We sincerely thank the reviewer for the encouraging and detailed feedback. Here we list our responses to the weaknesses and questions suggested by the reviewer.
>
> **W1: Selection of $\lambda$.**
>
> The additional parameter $\lambda$ is introduced to model the decision maker’s confidence in using the prior knowledge $\theta$ as a proxy for the parameter of interest $\beta$. Similar to how the budget constraint $\delta$ is specified, $\lambda$ can be selected using data-driven methods, such as grid-search cross-validation over the pair $(\delta, \lambda)$, as demonstrated in Simulation 2 of Section 4.2 (Line 372, Right).
>
> **Q1: Unhelpful $\theta$ and Convergence.**
>
> Let $\beta^*$ denote the solution to the stochastic optimization (SO) problem in the target domain (Line 125, Right). When the correlation between $\beta^*$ and $\theta$ is small, we can employ the weak-transferring mechanism developed in Section 3.3.2 (Line 250, Right). In such cases, the data-driven selection of $\lambda$ is expected to yield small values, effectively reducing the KG-WDRO problem to a nearly standard DRO formulation, thereby not hampering the learning. This intuition on the positive relationship between the informativeness of the prior knowledge and the size of $\lambda$—is demonstrated in the figure (https://figshare.com/s/05b069e330136338fa4e) through the ablation study on $\lambda$ in Section **W3** of our rebuttal to Reviewer LRuU.
>
> For sufficiently large datasets, following the approach in (Blanchet et al, 2022) (Line 476, Left), we can select the Wasserstein ball radius on the order of $n^{-1}$, i.e., $\delta_n = C/n$ for some constant $C > 0$. This ensures that the KG-WDRO solution, $\beta_{\text{KG-DRO}}$, converges to the optimum $\beta^*$ at an optimal rate of $O(n^{-1/2})$. Although this result has not been formally proven, we aim to establish it in future work. Notably, this $O(n^{-1/2})$ convergence rate holds for all $\lambda \in [0,\infty]$, including $\lambda=\infty$.
>
> **Q2: Optimization and Transfer Learning when $\lambda = \infty$.**
>
> When $\lambda = \infty$, our formulation is equivalent to adding equality constraints on the support of the perturbation. Specifically, if we define the perturbation as $\Delta = x - x'$ where $x’$ is the perturbed value of $x$, then the constraint $\Delta^\top \theta = 0$ must hold. This does not restrict the learning problem to a constraint optimization but rather constrains the support of the perturbation. Moreover, we do not think that transfer learning contradicts optimization problems. While our goal is to perform transfer learning—leveraging prior knowledge for the target domain—our computational method is based on solving optimization problems. To draw an analogy, in deep learning, the objective might be classification, yet the method relies on minimizing loss functions through gradient descent or similar optimization techniques.

---

### Decision · Program_Chairs · 2025-05-01

**Decision:**

Accept (poster)

**Comment:**

The paper presents a novel extension of Wasserstein-based DRO approaches that incorporates external knowledge into the ambiguity set via a modified cost function. By adjusting the Wasserstein distance, the framework aims to reduce the inherent conservatism of standard approaches. The authors demonstrate that the resulting optimization is equivalent to known regularization techniques and support their claims through theoretical analysis and numerical results.

Most of the reviewers' assessments for the paper are positive, and the overall contribution is conceptually solid, with clear potential for influencing future work in robust learning methods. Even though the paper has some limitations such as the lack of performance guarantees, the usage of prior knowledge to constraint the ambiguity set seems both novel and useful. I would suggest the authors to update the manuscript following the comments provided by the reviewers.